# Tuning the many-body interactions in a helical Luttinger liquid

Junxiang Jia[1], Elizabeth Marcellina[1], Anirban Das [2,3], Michael S. Lodge[1], BaoKai Wang[4], Duc-Quan Ho[1], Riddhi Biswas[1], Tuan Anh Pham[1], Wei Tao[1], Cheng-Yi Huang[4], Hsin Lin [5], Arun Bansil [4], Shantanu Mukherjee[2,3,6] & Bent Weber [1,7] ✉

In one-dimensional (1D) systems, electronic interactions lead to a breakdown of Fermi liquid theory and the formation of a Tomonaga-Luttinger Liquid (TLL). The strength of its many-body correlations can be quantified by a single dimensionless parameter, the Luttinger parameter $K$, characterising the competition between the electrons' kinetic and electrostatic energies. Recently, signatures of a TLL have been reported for the topological edge states of quantum spin Hall (QSH) insulators, strictly 1D electronic structures with linear (Dirac) dispersion and spin-momentum locking. Here we show that the many-body interactions in such helical Luttinger Liquid can be effectively controlled by the edge state's dielectric environment. This is reflected in a tunability of the Luttinger parameter $K$, distinct on different edges of the crystal, and extracted to high accuracy from the statistics of tunnelling spectra at tens of tunnelling points. The interplay of topology and many-body correlations in 1D helical systems has been suggested as a potential avenue towards realising non-Abelian parafermions.

A one-dimensional (1D) electronic system is described as a Tomonaga-Luttinger liquid (TLL) rather than a Landau-Fermi liquid, since the strictly 1D nature of the Fermi surface with only two Fermi points ($\pm k_F$) leads to a breakdown of screening[1]. As a consequence, electronic correlations are strong, even for weak electron–electron interactions, leading to collective excitations[2]. The elementary excitations of a TLL can be described as bosons, rather than fermions, and are given by charge and spin density waves with spin-charge separation[1,3]. For weak interactions, the correlation strength can be approximated by a single dimensionless parameter, the Luttinger parameter[4],

$$K \approx \left[ 1 + \frac{W(q=0)}{\pi \hbar v_F} \right]^{-1/2}. \qquad (1)$$

Here, $\pi \hbar v_F$ is the inverse density of states for 1D Dirac electrons, with a constant Fermi velocity $v_F$, and $W(q=0)$ is the energy of electrostatic interactions in the long wavelength limit. The former depends on the detail of the electronic band dispersion directly, while the latter can be expected to be susceptible to the dielectric environment of the 1D charge distribution, screening the many-body interactions. The Luttinger parameter can thus be expected to be highly tunable, ranging within its theoretical bounds, $0 < K < 1$, for repulsive (e.g. Coulombic) interactions, where $K = 1$ [$W(q=0) = 0$] represents the limit of non-interacting electrons. A threshold $K \approx 1/2$ is naturally given where the Coulomb and kinetic energy terms become equal, thus defining $K < 1/2$ as the limit of strong interactions.

Luttinger liquids have been detected in several realisations of 1D electronic structure, including linear chain organic conductors[5],

[1]Division of Physics and Applied Physics, School of Physical and Mathematical Sciences, Nanyang Technological University, Singapore 637371, Singapore. [2]Department of Physics, Indian Institute Of Technology Madras, Chennai, Tamil Nadu 600036, India. [3]Center for Atomistic Modelling and Materials Design, Indian Institute Of Technology Madras, Chennai, Tamil Nadu 600036, India. [4]Department of Physics, Northeastern University, Boston, MA 02115, USA. [5]Institute of Physics, Academia Sinica, Taipei 115201, Taiwan. [6]Quantum Centre for Diamond and Emergent Materials, Indian Institute of Technology Madras, Chennai, Tamil Nadu 600036, India. [7]ARC Centre of Excellence for Future Low-Energy Electronics Technologies (FLEET), School of Physics & Astronomy, Monash University, Clayton, VIC 3800, Australia. ✉e-mail: b.weber@ntu.edu.sg

metallic single-wall carbon nanotubes (SWNTs)[4,6], quantum Hall edge channels[7], cold atoms[8], atomic wires[9], and even crystalline mirror-twin boundaries in semiconducting transition metal dichalcogenides[10].

A key experimental signature of a TLL is a pseudogap suppression in the local density of states (LDOS), measured via tunnelling into the 1D electronic system, e.g., from the tip of a scanning tunnelling microscope (STM). Such pseudogap suppression is often referred to as zero-bias anomaly (ZBA) as it is strictly centred at the Fermi energy $E_F$[6,9–11]. In a TLL, the tunnelling conductivity around such ZBA scales universally, that is, as a power law in both bias voltage and temperature, with the same scaling exponent $\alpha$, related to the Luttinger parameter as $\alpha = C\left(K + K^{-1} - 2\right)$[12]. Different from the spin-degenerate (spinful) parabolic dispersions of conventional 1D metallic systems ($C = 1/4$), quantum spin Hall (QSH) insulators host linearly dispersing 1D edge states, in which the spin polarity is locked to the crystal momentum (helicity)[13,14]. For a helical Luttinger Liquid, strong spin orbit coupling causes the spin and chirality indices to coincide, and the bosonized Hamiltonian can be expressed as an effectively "spinless" Luttinger liquid with $C = 1/2$. TLL formation in such 1D helical systems is not only of fundamental interest as a testbed to study the interplay of topology and strong correlations, but may further constitute a potential host platform to realise non-Abelian parafermions[15–19].

Tuning of the many-body interaction strength has so far been demonstrated only for spinful Luttinger liquids, for instance, by applied gate voltages[20,21], or magnetic fields[22]. In this work, we demonstrate control of the Luttinger parameter $K$ in a helical TLL at the edges of the quantum spin Hall insulator 1T'-WTe$_2$, in which $K$ is found distinct for different crystal edges, and can be controlled by dielectric screening via the substrate. This becomes evident from a dependence of the Luttinger parameter on substrate dielectric constant and crystal edge, ranging between $K = (0.21 \pm 0.01)$ and $K = (0.33 \pm 0.01)$ in the

strong coupling limit. Our conclusions are based on temperature-dependent scanning tunnelling spectroscopy, demonstrating universal scaling, combined with a detailed statistical analysis of tens of tunnelling points along WTe$_2$ edge states, showing that values of $K$ fall into normal distributions of distinct statistical mean for different substrates and crystal edges.

The experimental extraction of $K$ is further supported by both a numerical model to estimate the strength of screened Coulomb interactions for the edge state's charge distribution and a material-specific real-space tight-binding model, which takes charge distribution and WTe$_2$ edge dispersion self-consistently into account. Both models are able to reproduce the trend of the Luttinger parameter qualitatively in different screening environments. Interestingly, despite being in the strong coupling limit ($K < 0.5$), where deviations from Eq. (1) may be expected due to strong interactions and a renormalised Fermi velocity, we show that the perturbative approximation can allow for qualitative predictions of its fundamental dependencies in terms of the Coulomb interaction strength and edge state dispersion. In the limit of very strong interactions ($K < 1/4$) reached in some of our samples, two-particle back-scattering of helical electrons may become significant[23,24], potentially leading to additional effects such as charge density wave formation, or even fractional Wigner crystallisation[24]. Eq. (1) can thus serve as an important indicator for the strength and tunability of Coulomb correlations in a Tomonaga–Luttinger liquid.

## Results

Helical TLLs have been reported for the 1D topological edge states of InAs/GaSb semiconductor heterostructures[23] and, more recently, for the edge states of atomically flat bismuthene[11]. An alternative realisation of an atomically thin QSH insulator[18,19] is 1T'-WTe$_2$[15,25–29], with

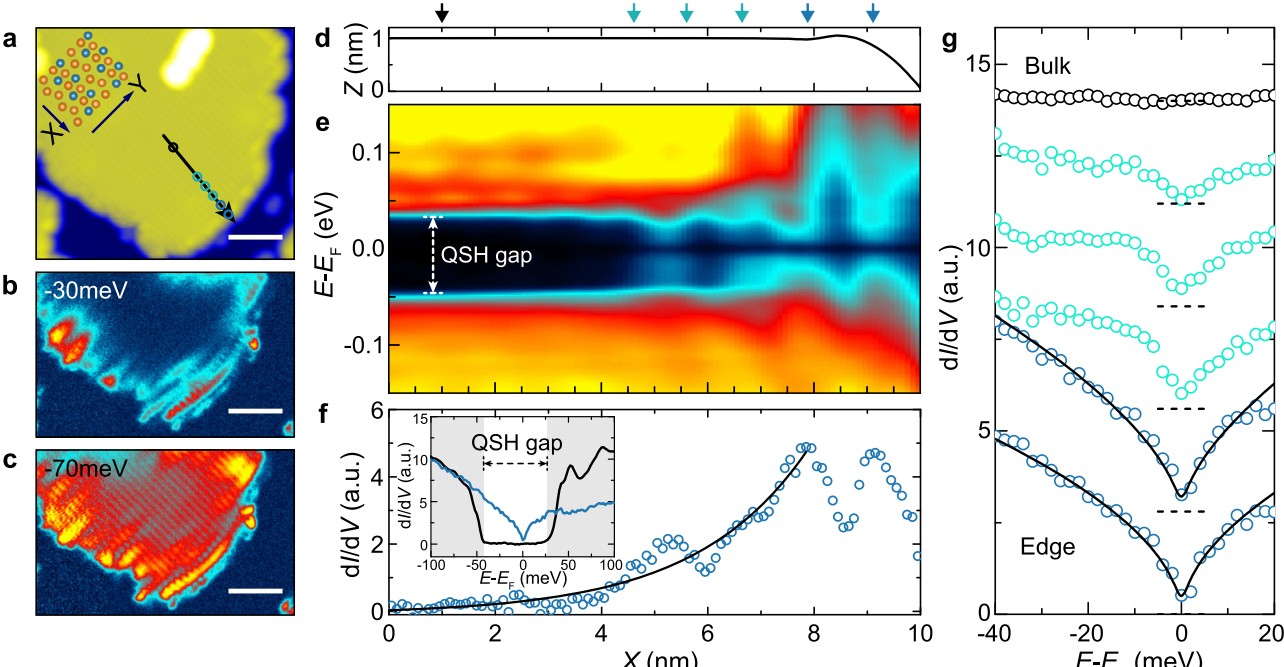

**Fig. 1 | Edge states in monolayer 1T'-WTe$_2$. a** STM topograph of a monolayer 1T'-WTe$_2$ crystal (scale bar: 5 nm). Insert: Crystal structure of the 1T'-WTe$_2$ crystal structure with W atoms (blue) and Te atoms (orange). Arrows represent different edge terminations. **b, c** Corresponding differential conductance (d$I$/d$V$) maps of the same island as in **a**, measured in constant height mode. **d–f** Spatial profile of the measured local density of state (LDOS) at 4.5 K, across the monolayer edge along the black arrow in **a**, and compared to a corresponding STM height profile (**d**). We extract a gap of -70 meV in the 2D bulk. The data in **f** has been obtained by integration of the LDOS in **e** between $E$ = −20 meV to $E$ = −40 meV and shows an

overall enhancement near the edge with superimposed charge density modulations also observed in **b**. The solid black line shows a fit to an exponential decay from which we extract a decay length of (2.1 ± 0.2) nm. Insert: Comparison of point spectra at bulk and edge, highlighting the bulk gap (black) and LDOS enhancement at the edge (blue) corresponding to the edge state. **g** Evolution of the edge state LDOS from the edge into the bulk (see corresponding markers in **a, d**). The solid black lines are fits to TLL theory (see text). All spectra have been offset for clarity, with the zero position indicated (horizontal dashed lines).

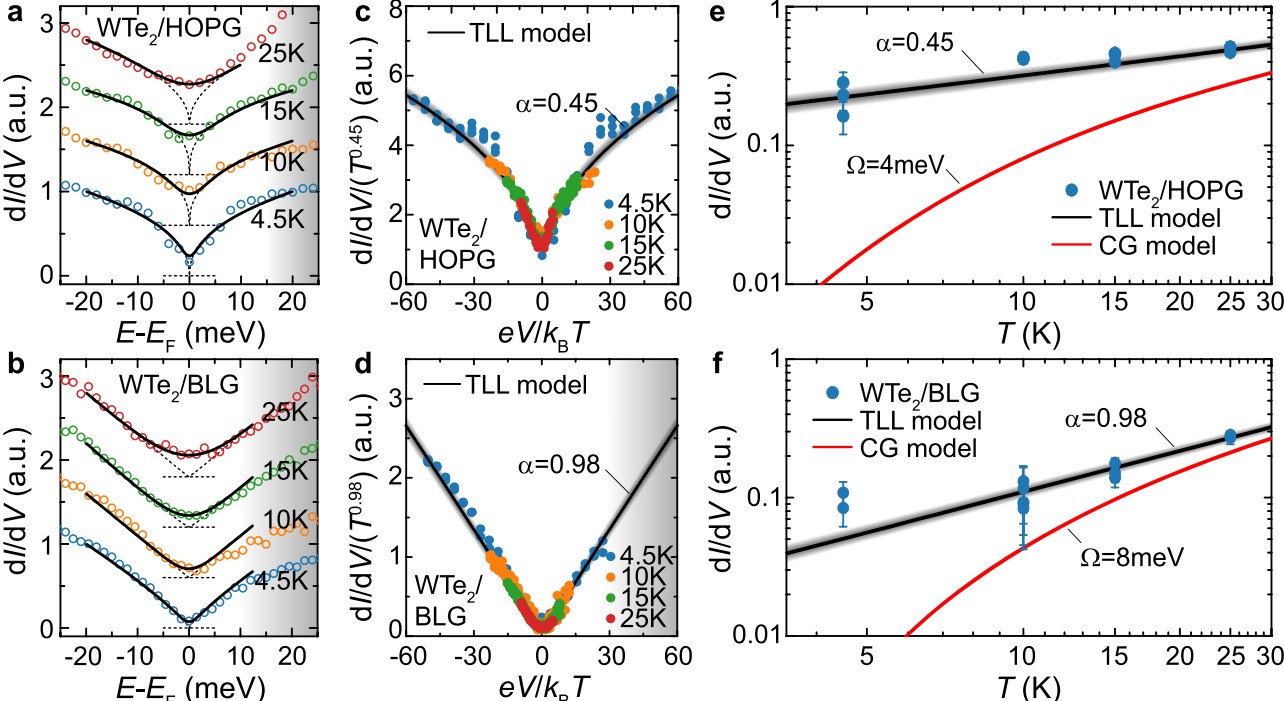

**Fig. 2 | Demonstration of universal scaling for tunnelling into a Tomonaga-Luttinger Liquid (TLL). a, b** Temperature dependent point spectra of the edge state's local density of states (LDOS) up to $T = 25$ K, measured at the Y-edges off $WTe_2$/HOPG and $WTe_2$/BLG, respectively. The tunnelling spectra are offset by 0.6 on the y-axis with the zero position highlighted by horizontal dashed lines. Solid black lines are fits to TLL theory, taking into account thermal broadening (Section S3 of the Supplementary Information). The corresponding bare power-law fits are shown as dotted curves. The shaded bands indicate the conduction band tail. **c, d** The data in **a** and **b** can be collapsed onto a single universal curve with common scaling exponents $\alpha = 0.45$ ($WTe_2$/HOPG) and $\alpha = 0.98$ ($WTe_2$/BLG). Shaded bands indicate statistical confidence defined by the mean absolute error across the individual fits at different measurement temperature. The shaded band in **d** indicates the conduction band tail. **e, f** Temperature dependence of the measured zero-bias ($E = E_F$) conductance in **a**–**d** (blue circles), compared with the TLL model (solid black lines) and Coulomb gap (CG) model (solid red lines). Fitting parameters of the CG model, such as the disorder parameter $\Omega$, have been determined from a best fit to each data at $T = 4.5$ K (see Supplementary Fig. 4). The error bars are obtained from the Gaussian width of the TLL fitting residual. Shaded bands indicate statistical confidence as defined for (**c, d**).

topological band gaps of a few tens of meV[26,27], tunable by electric fields and strain. A typical 1T'-$WTe_2$ monolayer crystal is shown in Fig. 1a, synthesised by van-der-Waals molecular-beam epitaxy (MBE)[26,27] on highly oriented pyrolytic graphite (HOPG) (see Methods). We achieve typical island circumferences $L = 100$–400 nm with disordered edges. Locally, however, we find straight and well-defined sections, with typical lengths between 10–20 nm that are terminated either perpendicular (Y-edge) or parallel (X-edge) to the atomic rows.

Corresponding differential conductance (d$I$/d$V$) maps of the same area are shown in Fig. 1b, c, reflecting the expected QSH electronic signature of an enhanced local density of states (LDOS) at the crystal edge around an insulating 2D bulk[26,30]. A larger area of the same crystal can be viewed in Supplementary Fig. 1, showing that the electronic edge state is largely continuous along the island's circumference, even in the presence of local edge roughness and disorder, and regardless of edge direction or termination, and at adjoining corners. The relevant length-scale determining the electrostatic energy of the screened Coulomb interactions within the edge state's 1D charge distribution is therefore set by the islands' circumference. Superimposed spatial modulations in the charge density can be shown to arise from edge roughness even in otherwise perfect monolayer crystals (Supplementary Fig. 7). However, as these modulations –more clearly pronounced on the X-edges –appear quasi-periodic, additional mechanisms may be at play, including spatial inhomogeneity of the Fermi velocity and/or interaction-mediated scattering of the edge state, leading to charge density wave order.

We note that we do observe semi-metallic behaviour and a variety of in-gap states in some monolayer crystals, especially in the vicinity of

lattice defects and adsorbates (Supplementary Fig. 1). Some crystals also show a small but finite 2D LDOS within the bulk and slight variations in gap size. For this study, however, we focus on larger clean regions, such as shown in Fig. 1a–c, in which a well-defined suppression of the bulk LDOS is observed.

A spatial profile of the local density of states (LDOS), measured along the black arrow in Fig. 1a, is shown in Fig. 1e, alongside a corresponding STM height profile (Fig. 1d). The data confirms a well-defined energy gap over ~70 meV in the 2D bulk, similar in magnitude to earlier reports[26,27], and in agreement with the tight-binding model adopted in this work[31] (compare Fig. 3d, e). The gap is seen to narrow and eventually disappear at the edge, consistent with the presence of a metallic edge state. We identify the position of the edge, electronically, from a profile of the energy-integrated LDOS (−20 to −40 meV) in Fig. 1f, which shows a finite width $\Delta x \approx 2$ nm with exponential decay into the 2D bulk. From an exponential fit, we extract a decay length of $(2.1 \pm 0.2)$ nm, significantly larger compared to the crystal lattice constant ($a = 0.348$ nm). Spatial modulations superimposed onto the overall edge state LDOS enhancement further indicate that the observed LDOS enhancement is not due to localised atomic orbitals, but rather are an electronic effect due to the metallic edge state.

The difference between bulk and edge state LDOS is further highlighted by the direct comparison of individual STS point spectra shown in the insert to Fig. 1f. For a linear (Dirac) dispersion in 1D, one would expect a constant (energy-independent) edge state LDOS throughout the gap. Instead, we observe a clear V-shaped suppression with a minimum at the Fermi energy $E = E_F$ (ZBA). This ZBA – a first indication of the presence of 1D correlations[11,26,30]– is strictly confined

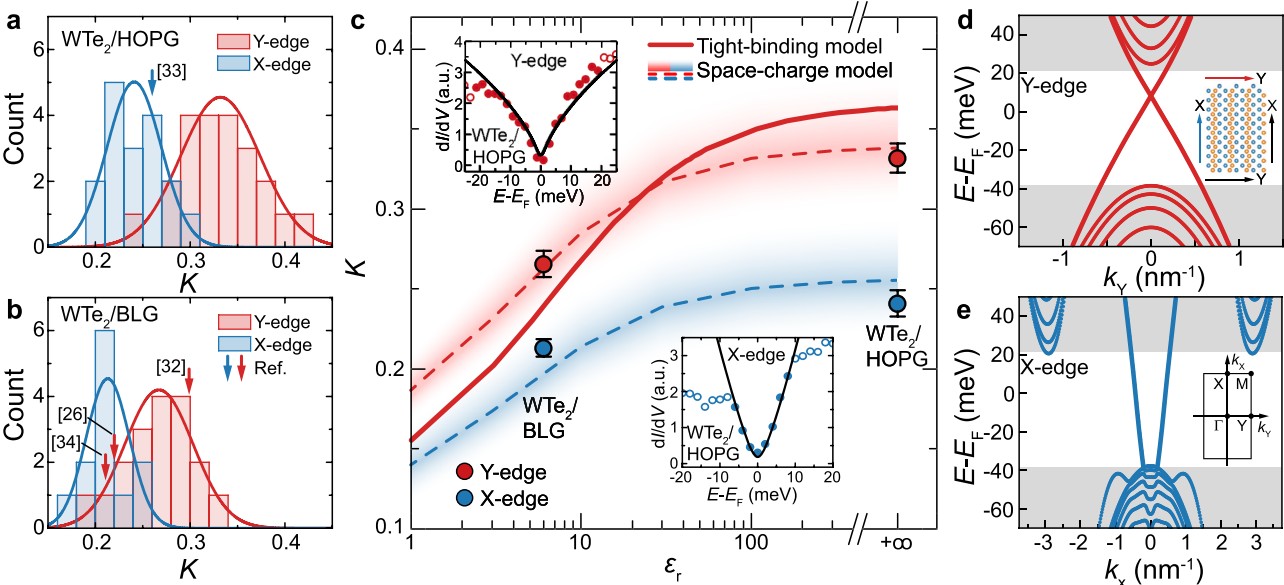

**Fig. 3 | Dielectric screening of the TLL many-body interactions. a, b** Histograms of the Luttinger parameter $K$, recorded across 69 locations along the X- and Y-edges of WTe$_2$/HOPG and WTe$_2$/BLG islands. Solid lines are fits to Gaussian normal distributions characterised to extract statistical mean and standard deviation. Arrows indicate data points from literature[26,33–35]. **c** Luttinger parameter $K$, extracted across different samples, and literature, plotted as a function of the substrate dielectric constant $\epsilon_r$. Red (blue) circles are the mean and standard error of the mean extracted for the distributions in **a, b**. Shaded bands and dashed lines are numerical calculations of $K$ based on the electrostatic energy $W(q = 0)$ stored in a screened edge charge distribution of varying density (see text), as well as the calculated edge dependent Fermi velocity $v_F$ extracted from **d** and **e**. The bands have been modeled to coincide in width with the extracted standard deviation of the distributions in a and b. The solid red line represents tight-binding calculations of the Luttinger parameter. Insert: Representative spectra measured at the Y- and X-edge, respectively, on HOPG, with extracted Luttinger parameters $K \approx 0.35$ (Y-edge) and $K \approx 0.24$ (X-edge) from TLL fits shown as black lines. For the dielectric constants, we assume $\epsilon_r = 6$ (BLG/SiC)[40] and $\epsilon_r \rightarrow \infty$ (HOPG), approximating the perfect metal limit. **d, e** Tight-binding band structure of the Γ-Y and Γ-X directions, respectively, assuming the atomic terminations shown by coloured arrows in (insert to **d**). Grey shading indicates the 2D bulk band edges with a band gap comparable to the data in Fig. 1e, f. Calculations are based on the edge terminations highlighted in the insert to (**d**), showing the position of W atoms (blue) and Te atoms (orange). Insert to (**e**), Corresponding WTe$_2$ Brillouin zone.

to the crystal edges as confirmed in Fig. 1g, where we plot individual d$I$/d$V$ point spectra away from the Y-edge into the monolayer bulk. Black solid lines are fits to the theoretically predicted tunnelling density of states into a TLL at 4.5 K, following the universal scaling relationship[4],

$$\rho_{TLL}(E, T) \propto T^\alpha \cosh\left(\frac{E}{2k_BT}\right)\left|\Gamma\left(\frac{1+\alpha}{2} + i\frac{E}{2\pi k_BT}\right)\right|^2 \quad (2)$$

Here, $\alpha$ is the scaling exponent, $E - E_F \equiv eV$ is the STM bias voltage, $T$ is temperature, and $\Gamma(x)$ is the gamma function. For our fits, we further take broadening of the LDOS into account, consistent with the procedure in Ref. 11 (as shown in Supplementary Fig. 2).

Particular to 1D helical systems, the linear (Dirac) dispersion supports the bosonisation picture of TLL theory over a much larger energy range. Contrasting the spinful parabolic dispersions of conventional metals, which deviate from a linear approximation already at moderately low energy, the 1T′-WTe$_2$ edge state dispersion remains linear over a wide energy range within the QSH band gap[31] (compare Fig. 3d, e), consistent with the range of our TLL fits (typically −20 meV to +15 meV). From 33 edge spectra (two of which are shown in Fig. 1g), we extract $\alpha = 0.59 \pm 0.02$ and $K = 0.35 \pm 0.02$, assuming helicity of the QSH edge state ($C = 1/2$). These numbers are in good agreement with values recently reported for the helical edges of bismuthene ($K \approx 0.42$)[11], which features similarly tightly confined 1D edge states, in contrast to the larger edge state decay lengths of semiconductor-based QSH systems[23].

We note that, in principle, a narrow suppression in the LDOS of 1D electronic systems[1,4,11,26,30] may have its origin in a number of different mechanisms, most notably the formation of a Coulomb gap (CG) due to long-range interactions[32]. However, as we show in Supplementary Fig. 4 and 5, a Coulomb gap (CG) model cannot explain the detailed

functional form of the ZBA observed. To further rule out the presence of a CG, we turn to temperature-dependent scanning probe spectroscopy, to confirm universal scaling of the TLL tunnelling conductance. In Fig. 2a, d, we compare d$I$/d$V$ spectra measured on WTe$_2$/HOPG with those taken on WTe$_2$/BLG. We limit the temperature range explored to $T < 25$ K, as higher temperatures are found to give rise to a significant bulk contribution to the overall LDOS (Supplementary Fig. 5e). Black solid lines in Fig. 2a, b are fits to the universal scaling relationship (Eq. (2)), yielding comparable power-law exponents $\alpha \approx 0.45$ (HOPG) and $\alpha \approx 0.98$ (BLG) at all temperatures. The corresponding bare power law is indicated by dotted lines. Universal scaling is confirmed by normalising the measured d$I$/d$V$ with $T^\alpha$ and plotting the result as a function of the dimensionless energy parameter $eV/k_BT$. Following this procedure, all spectra collapse onto a single universal curve (Fig. 2c, d) with power-law exponents $\alpha = 0.45 \pm 0.03$ ($K = 0.40 \pm 0.01$) for WTe$_2$/HOPG and $\alpha = 0.98 \pm 0.03$ ($K = 0.27 \pm 0.01$) for WTe$_2$/BLG. The uncertainty in the fitted values of $\alpha$, indicated by the shaded confidence bands in the figure, arises from the mean absolute error of TLL fits across the different measurement temperatures. As a further confirmation of universal scaling, we plot the zero-bias ($E = E_F$) tunnelling conductance as a function of temperature in Fig. 2e, f, comparing the expected temperature dependencies of TLL and CG theories, respectively. The temperature dependence of the TLL is in excellent agreement with the universal scaling exponents extracted from Fig. 2a–d, while the CG model shows substantial deviations from the measured data on both substrates, especially at temperatures <10 K.

Universal scaling as demonstrated is usually considered sufficient proof for the presence of a TLL[11]. Further to this, however, below we show that the observed differences in the Luttinger parameter across substrates can be explained by substrate-induced screening of the

many-body interactions. To this end, we have performed a detailed statistical analysis of $K$ values extracted from a total of 69 locations, measured along the edge states of crystals on both substrates, plotted in the histograms of Fig. 3a, b. We find that $K$ follows normal distributions with distinct statistical mean but comparable standard deviation for both edge terminations and substrates. This is further confirmed by fits to Gaussian normal distributions (solid lines), from which we extract $K = 0.27 \pm 0.01$ (Y-edge) and $K = 0.21 \pm 0.01$ (X-edge) for WTe$_2$/BLG and $K = 0.33 \pm 0.01$ (Y-edge) and $K = 0.24 \pm 0.01$ (X-edge) for WTe$_2$/HOPG. The uncertainty is given by the standard error of the mean $\sigma/\sqrt{n}$ which quantifies the accuracy of the extracted mean, reflecting that $K$ was obtained from a population of $n$ sample points with standard deviation $\sigma$. Further detail on the statistical analysis, including normality tests, and statistical significance of the differences in the mean is given in Supplementary Table 1. Finally, we include $K$ values extracted from edge state tunnelling spectra in literature[26,33–35], and find that their values fall within the same distributions as shown in Fig. 3a, b.

The mean and standard error of the mean for both edges are plotted as a function of the substrate dielectric constant $\epsilon_r$ in Fig. 3c. Across all data, the Luttinger parameter $K$ increases at low $\epsilon_r$ and saturates for $\epsilon_r \to \infty$, where the latter corresponds to HOPG[36]. We note that although there have also been reports of a finite dielectric constant for HOPG ($\epsilon_r \approx 53$)[37], $K$ starts to saturate at ($\epsilon_r \approx 50$), implying that $\epsilon_r \to \infty$ provides a reasonable approximation in this context.

To explain this trend, we devise a qualitative model for $K$[38], based on the perturbative approximation (Eq. 1) and the assumption of a substrate-dependent mirror charge, screening many-body Coulomb interactions within the helical edge. In our model, edge state and mirror charge are approximated by space-charge distributions $\rho(x, y, z)$ and $\rho'(x, y, z) = -\frac{\epsilon_r - 1}{\epsilon_r + 1}\rho(x, y, -z)$, respectively, allowing us to calculate the electrostatic energy $W(q = 0)$ of the total charge numerically (see Section S6 of the Supplementary Information). In order to estimate $K$ for the respective Y- (X-) edges, we extract the Fermi velocities $v_F = 1.16 \times 10^5$ m/s ($2.69 \times 10^5$ m/s) from the tight-binding dispersions in Fig. 3b–e. The red and blue shaded bands in Fig. 3c have been calculated for an average edge length $L = 200$ nm, and space-charge densities $\rho_{0,y} = 0.0039 \pm 0.0007$ e/nm$^3$ and $\rho_{0,x} = 0.0082 \pm 0.0009$ e/nm$^3$ to reflect the standard deviations consistent with Fig. 3a, b. The observed trend is further confirmed by our material-specific tight-binding model, taking edge state charge density and band dispersion self-consistently into account (Supplementary Fig. 7). The solid red line in Fig. 3a represents the calculated the Luttinger parameter $K$ along the Y-edge.

The ability of our model to reproduce the trends in $K$ may seem surprising at first, given that Eq. (1) should strictly only apply in the perturbative limit of weak interactions ($K > 0.5$). However, we note that several factors can extend the range of validity of Eq. (1), especially in helical systems. For instance, Coulomb interactions do not allow for backscattering with large momentum transfer $q = 2k_F$ in a helical TLL. This supports a perturbative formalism which primarily includes the long wavelength forward scattering modes ($q \approx 0$). More so, our observation of a finite width TLL in which the LDOS falls exponentially from the edge into the QSH bulk (compare Fig. 1f) leads to a Coulomb interaction extending up to the fifth nearest neighbour ion of the lattice (Supplementary Fig. 8). This justifies the approximation $g_2 = g_4$ implied in our perturbative analysis, where $g_2$ and $g_4$ correspond to interaction terms in a Hamiltonian contribution, $g_4(\psi_{R\uparrow}^\dagger \psi_{R\uparrow})^2 + g_2(\psi_{R\uparrow}^\dagger \psi_{R\uparrow} \psi_{L\downarrow}^\dagger \psi_{L\downarrow})$, where $\psi_{R\uparrow}$ annihilates a right mover electron with spin up, and $\psi_{L\downarrow}$ annihilates a left mover electron with spin down. We conclude that while the precise quantitative match of $K$ may be fortuitous, the perturbative treatment is able to correctly predict the trend in $K$, for different edge dispersions and substrate mediated screening. This work therefore presents a precision experimental determination of the effect of kinetic and potential energy terms on the Luttinger parameter, and may form a benchmark for future theoretical studies.

To summarise, we have demonstrated tunability of the many-body interactions in a Tomonaga-Luttinger Liquid (TLL) at the helical edges of the quantum spin Hall insulator 1T'-WTe$_2$. Synthesising WTe$_2$ on metallic van-der-Waals substrates, we have been able to demonstrate effective screening of the many-body Coulomb interactions, reflected in a significant enhancement of the Luttinger parameter. The cleanest signatures of the TLL are observed on the well-screened Y-edges of WTe$_2$/HOPG, where $K = (0.33 \pm 0.01)$. Here, the higher degree of tunability observed would suggest that systems of lower Fermi velocity and low carrier density may allow to explore a wider range of $K$, something which may be confirmed in other QSH material systems. In turn, the strongest many-body interactions are found on the unscreened X-edges for WTe$_2$/BLG where we measure $K = (0.21 \pm 0.01)$ – the lowest reported for any helical TLL system to date. Indeed, this value represents the limit of very strong interactions ($K < 0.25$), where additional mechanisms, such as charge density wave formation due to two-particle scattering or even Wigner crystallisation[24] may be expected. Indeed, the quasi-periodic charge density modulations observed on X-edges are not inconsistent with such phenomena and will be a focus for future investigation into the role of edge disorder and scattering. We thus expect that our results will stimulate further experimental and theoretical investigations of the interplay of topology and strong electronic interactions in 1D, also in the superconducting state, in which non-Abelian parafermions are predicted[18,39].

## Methods

### MBE growth

Monolayer 1T'-WTe$_2$ was synthesised by molecular-beam epitaxy (MBE) on highly oriented pyrolytic graphite (HOPG) and bilayer graphene (BLG) grown on 6H-SiC(0001) substrates in an Omicron Lab10 ultrahigh vacuum (UHV) MBE chamber with a base pressure of $2 \times 10^{-10}$ mbar. Prior to MBE, the HOPG substrates were mechanically cleaved in ambient conditions and then degassed at 300 °C overnight. The 6H-SiC(0001) substrates were degassed at 600 °C overnight in UHV and then flash annealed at 2000 °C for a few cycles to graphitise producing BLG. The WTe$_2$ islands were grown by evaporating high-purity W(99.998%) and Te(99.999%) with a flux ratio of 1:280 and substrate temperature of 165 °C for 2 h to cover approximately 50% of the monolayer area. In total, we measured three 1T'-WTe$_2$/HOPG samples and another three of 1T'-WTe$_2$/BLG.

### Scanning tunnelling spectroscopy

Low-temperature scanning tunnelling microscopy and spectroscopy (STM/STS) measurements were performed in an Omicron low-temperature STM ($\simeq 4.5$ K) under UHV conditions ($\simeq 5 \times 10^{-11}$ mbar). For temperature-dependent STM/STS, a heater element fitted to the sample stage was used for counter-heating from 4.5 K. For all spectroscopy measurements, we used a platinum/iridium tip calibrated against the Au(111) Shockley surface state. The spectroscopy measurements were carried out using standard lock-in techniques with a modulation amplitude of $V_{ac} = 2$ mV and a modulation frequency of 1.331 kHz. The value of $V_{ac}$ was optimised to maximize the signal-to-noise ratio while eliminating instrumental broadening (Supplementary Fig. 3). Differential conductance maps were taken in constant height mode.

## Data availability

The scanning tunnelling spectroscopy data generated in this study have been deposited in the NTU research data repository DR-NTU (Data) database at https://doi.org/10.21979/N9/J8BWVQ.

## Code availability

The computer code used for data analysis and modelling is available from BW or SM upon reasonable request.

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

## Acknowledgements

This research is supported by the National Research Foundation (NRF) Singapore, under the Competitive Research Programme "Towards On-Chip Topological Quantum Devices" (NRF-CRP21-2018-0001) (BW) with further support from the Singapore Ministry of Education (MOE) Academic Research Fund Tier 3 grant (MOE2018-T3-1-002) "Geometrical Quantum Materials" (BW). The work at Northeastern University was supported by the US Department of Energy (DOE), Office of Science, Basic Energy Sciences Grant No. DE-SC0022216 and benefited from Northeastern University's Advanced Scientific Computation Center and the Discovery Cluster and the National Energy Research Scientific Computing Center through DOE Grant No. DE-AC02-05CH11231 (AB). HL acknowledges the support by the Ministry of Science and Technology (MOST) in Taiwan under grant number MOST 109-2112-M-001-014-MY3. SM acknowledges financial support from SERB grant number SP/2021/1008/PH/SERB/008839. BW acknowledges a Singapore National Research Foundation (NRF) Fellowship (NRF-NRFF2017-11). We thank T. Giamarchi, R. Claessen, M.S. Fuhrer, and J. Klinovaja for invaluable discussions.

## Author contributions

JJ, DQH, WT, and TAP grew the WTe$_2$ monolayers. JJ, DQH, and MSL performed the scanning tunnelling spectroscopy experiments. AD, JJ,

RB, CYH, BKW, HL, AB and SM performed the theoretical calculations. JJ, EM, and BW analyzed the data. HL and SM supervised the theory work. BW conceived and coordinated the project. JJ, EM, SM and BW wrote the manuscript with input from all authors.

## Competing interests

The authors declare no competing interests.
