## [Peer Review File · Nature Communications]

REVIEWER COMMENTS

Reviewer #1 (Remarks to the Author):

The manuscript by Jia et al. reports the scanning tunneling microscopy (STM) measurements on the conducting edges of the monolayer 1T'-WTe₂ grown on two different substrates (highly oriented pyrolytic graphite (HOPG), and bilayer graphene (BLG)/SiC). The monolayer 1T'-WTe₂ is an exotic 2D monolayer, which combines topology, superconductivity, and strong correlations, each of which is a hot topic in condensed matter physics independently.

With systematic measurements, this work tried to convey that the conducting edge of the monolayer 1T'-WTe₂ shows consistent behaviors as a Tomonaga-Luttinger liquid (TLL), a one-dimensional metal with strong electron-electron interaction, and the interaction could be tuned by the bottom dielectrics (HOPG and BLG/SiC).

In my point of view, the manuscript is well-written, covering a broad background introduction, a systematic study of the edge orientations, and different dielectrics. The data presentation is technically sound, and notably, the systematic study of X and Y edge, including the histogram of the interaction strength K significantly enhanced the reliability of the conclusions. I would rate this work as a timely, high-impact study, which is worth publishing on Nature Communications if the following questions are well addressed.

1. It is not clear to me why C is $1/2$ considering the helicity, in the relation for experimentally extracted power-law α and theoretical Luttinger parameter K . The authors are expected to explain this in more detail. I understand some noted in the cited works, but it is important to explain them clearly in this work.
2. What is the definition of the strong coupling (the manuscript referred to $K < 1/2$), weak coupling (the manuscript referred to $K > 1/2$), and very strong interaction regime (the manuscript referred to $K < 1/4$)? How is the number chosen? It is confusing for general readers on the definition. Please explain it in a little more detail.
3. How does the Luttinger parameter K (or power-law) change with a distance to the edge? Fig. 1g only fits the K and zero-bias anomaly to those points at the author-defined edge, but not to those close to the "edge", or slightly from the edge. I am wondering if the universal scaling is still valid for those areas and whether the universal scaling is still valid.
4. The work presents the dielectric effect on the TLL and demonstrates a good fit for calculation, which is important to understanding the screening effect on the helical Luttinger liquids. However, there are only two data points of the ϵ_r . It is optional to get it better by adding one more datapoint if the authors have more data on other dielectrics, e.g., few-layer graphite or monolayer graphene. I fully understand the challenging in experiments, but this will make the work much higher standard in presentation and demonstration.
5. Since K is different for X and Y edges, a quick question is how the K will be at the intersection of the X and Y edges?

Other suggestions/comments that might be helpful:

1. In Ref. 10, the page number is wrong.
2. I would suggest replacing the ref. 13 (PRL 86,143 (2001)) with PRL 77, 253 (1996). The leading author is the same. This is the first demonstration of the TLL in fractional quantum Hall edges.

Reviewer #2 (Remarks to the Author):

Jia and collaborators study the possibility of controlling the Luttinger parameter K in a one-dimensional (1D) helical Tomonaga Luttinger Liquid (TLL) that forms along the edges of the quantum spin Hall insulator $1T'-WTe_2$. In a TLL, elementary excitations are replaced by collective modes. They consist of charge and spin density waves, with spin-charge separation, described as bosons rather than fermions. A single dimensionless parameter, the Luttinger parameter K , describes the correlation length of these collective modes. K depends on electrostatic interactions which are sensitive to the dielectric environment of the 1D TLL. Hence the idea to tune K by proximity to selected different substrates. To test this idea, Jia et al. study $1T'-WTe_2$ monolayer crystals synthesised on highly-oriented pyrolytic graphite and on bilayer graphene islands by molecular-beam epitaxy.

The experimental challenge is the identification of the TLL in the first place. Jia et al. rely on the suppression of the local density of states (DOS) at the Fermi level in the tunneling spectra measured using a scanning tunneling microscope -a standard scheme. They find that this so called zero-bias anomaly (ZBA) shows universal scaling behavior as a function of bias voltage and temperature. The lineshape of the ZBA, its perfect scaling with temperature, and the changing scaling factor for different substrates are all consistent with the presence of a TLL. These experimental results are based on systematic temperature-dependent scanning tunnelling spectra, and on a detailed statistical analysis of tens of tunneling points along $1T'-WTe_2$ edges. They show that K obeys normal distributions with distinct statistical mean for different substrates and crystal edges. These findings are corroborated both by numerical modelling to estimate the strength of the screened Coulomb interactions within the charge distribution along the edges, and by a material-specific real-space tight-binding model. The modelling further allows Jia et al to show that a Coulomb gap can neither account for the ZBA lineshape nor for its scaling.

Jia et al. propose a timely study with a very interesting result. The modelling appears reasonable, with a convincing fit of the ZBA and its scaling -although I am not fully proficient in the proposed theoretical modelling. On the other hand, these conclusions rely on experimental data that raise some questions and concerns I detail below. The authors need to address these issues before the work may be published.

A. How do the authors identify the X and Y edges? There are very few, if any, long straight edges in the images presented in Fig.1 and Fig.S1. Reading the text, one expects to find well defined and long X and Y edges (the authors talk about "100–400 micrometer long edges for some crystals) when the quoted length in fact refers to the perimeter of a monolayer grain. It would be more appropriate and significant to refer to the typical length of relevant edges.

B. It is often quite challenging to precisely identify the position of a step edge in topographic STM images. I am not convinced that Jia et al. have properly identified the edge in Figs.1d, 1e and 1f. The Z-profile in Fig.1d actually increases beyond the point identified as the edge, suggesting the monolayer edge might actually be up to 1nm further to the right. The proper identification will have an impact on the assignment of spectral features to an edge state, and on the fitting of the decay length in Fig.1f. Why would the STM tip height increase beyond the edge of the $1T'-WTe_2$ flake in a constant current scan?

C. The authors claim that the conductance maps of the monolayer $1T'-WTe_2$ flake in Figs.1b and 1c reflect "the expected QSH electronic signature of an enhanced local density of states (LDOS) at the crystal edge around an insulating 2D bulk". Looking at Fig.1 and Fig.S1, I find only one feature clearly compatible with a 1D edge state, namely along the lower right Y edge. The other edges, especially the X edges, show either no enhanced local DOS or elongated 1D-like features extending perpendicular rather than parallel to the edge. Moreover, the brighter 1D spectral feature along the lower right Y edge does not seem to sit exactly on the $1T'-WTe_2$ monolayer edge (see comment B). Why would there only be one edge with a parallel 1D DOS feature, and why do the authors identify features along the Y edges extending perpendicular into the 2D bulk as an edge state?

D. The statement that “An enhanced LDOS persists along the sample edge for all energies, but the bulk is insulating between approximately -60 meV and +50 meV, with the LDOS matching that of the HOPG substrate” in the legend of Fig.S1 is not compatible with the data it is supposed to describe. The bulk DOS is low and comparable to that of the HOPG substrate only between -40mV and +20mV bias. Above +20mV, it is larger than the DOS of HOPG, and even larger than the step edge DOS above +40mV.

E. The legend for panel g in Fig.1 is confusing. It says “Evolution of of (typo) the edge state LDOS from the edge into the bulk”. But the five dark blue spectra are all measured at a position identified by the authors as the step edge. Are they measured at the very same position or at different locations along the step edge? I suggest to really show the evolution of the spectra away from the step edge and show the reproducibility of the edge spectra in a separate panel.

Reviewer #3 (Remarks to the Author):

This manuscript presents STM study of the topological edge state in the quantum spin Hall insulator 1T'-WTe₂. The authors claim helical Luttinger liquid physics in these 1D channels by the observation of bias- and temperature-dependent tunneling spectra that show power-law behavior. While I find this topic interesting, I have some concerns that need to be clarified before I could recommend publication of the manuscript.

The topological edge state (TES) presented in this manuscript does not look to be a continuous 1D channel (Fig. 1 and Fig. S1). For example, in Fig. S1 it looks disconnected with lots of scatterers and breaks. Besides, the X-edge and the Y-edge show distinct Fermi velocities and completely different band shapes (Fig. 3d-e). Based on the above reasons the effective chain length of the 1D channels should be around 10 ~ 20 nm as can be seen in Fig. S1. This is because theory predicted that the impurity potential in a 1D TLL with repulsive interaction would renormalize to infinity in the low-energy limit, thus cutting a 1D TLL into shorter channels (e.g., chapter 27 in the book “Bosonization and Strongly Correlated Systems” by Alexander O. Gogolin et al.). For a $L \sim 20$ nm channel with a Fermi velocity $v_F \sim 1.16 \times 10^5$ m/s (this value is from the manuscript), the quantum confinement energy is about $E_0 = \hbar v_F \pi / L \sim 12$ meV, well above the experimental energy resolution. This should result in a hard gap at the Fermi level (e.g., Ref. 16 and arXiv: 2108.03829), rather than the V-shaped power-law gap observed in this manuscript. The authors need to carefully clarify this issue.

The authors claim that the TLL physics applies to the entire QSH gap range due to the linear dispersion of the TES. However, as can be seen in Fig. 2a-b, I do not see well power-law fits over the entire gap energy range, but instead fits over a smaller range (-20 meV to 15 meV) are shown. The experimental data also deviates from the power-law fit curves even within such a smaller energy range (e.g., 25 K in Fig. 2a and 10 K in Fig. 2b). In addition, the shaded area representing statistical confidence shown in Fig. 2c is quite broad $\sim 50\%$ (width is about 4 a.u. while the data point is about 8 a.u. at -30 eV/kBT). To support the authors' claim, the authors need to present better data or to refine their data analysis.

Detailed Responses to the Reviewers

We thank the reviewers for their positive assessment, and for the constructive remarks and criticism which, we believe, have further improved our manuscript. Following our amendments, with additional data, analysis, and theoretical calculations provided, we believe that the paper now meets the required quality of presentation and soundness required for Nature Communications.

Reviewer #1 (Remarks to the Author):

The manuscript by Jia et al. reports the scanning tunneling microscopy (STM) measurements on the conducting edges of the monolayer 1T'-WTe₂ grown on two different substrates (highly oriented pyrolytic graphite (HOPG), and bilayer graphene (BLG)/SiC). The monolayer 1T'-WTe₂ is an exotic 2D monolayer, which combines topology, superconductivity, and strong correlations, each of which is a hot topic in condensed matter physics independently. With systematic measurements, this work tried to convey that the conducting edge of the monolayer 1T'-WTe₂ shows consistent behaviors as a Tomonaga-Luttinger liquid (TLL), a one-dimensional metal with strong electron-electron interaction, and the interaction could be tuned by the bottom dielectrics (HOPG and BLG/SiC).

In my point of view, the manuscript is well-written, covering a broad background introduction, a systematic study of the edge orientations, and different dielectrics. The data presentation is technically sound, and notably, the systematic study of X and Y edge, including the histogram of the interaction strength K significantly enhanced the reliability of the conclusions. I would rate this work as a timely, high-impact study, which is worth publishing on Nature Communications if the following questions are well addressed.

We thank Reviewer #1, for the very positive assessment, highlighting that our study is “well-written” and “technically sound”, strengthened by our statistical analysis to present a “timely and high-impact study”. We are addressing the remaining queries in the details below.

1. It is not clear to me why C is 1/2 considering the helicity, in the relation for experimentally extracted power-law α and theoretical Luttinger parameter K. The authors are expected to explain this in more detail. I understand some noted in the cited works, but it is important to explain them clearly in this work.

We thank the reviewer allowing us to clarify. The pre-factor used ($C = \frac{1}{2}$) is consistent with previous demonstrations in quantum spin Hall insulators [1, 2], taking the 1D helical nature of the 1D QSH edge states into account (see Ref. [3] for scaling exponents of different TLL types). For the case of helical Luttinger Liquids (HLL), spin orbit coupling causes the spin and chirality indices to coincide, and the bosonized Hamiltonian can be expressed as an effectively “spinless” Luttinger liquid with simple replacement of the Luttinger parameter K with K_{HLL} ,

$$\alpha_{HLL} = \frac{1}{2}(K_{HLL} + K_{HLL}^{-1} - 2)$$

We note that a factor of 2 compared to definition in section III.B. of Ref. [3] arises from the expression of the density of states (Ref [2] main text) for tunnelling into a regular TLL (Eq. (61) of Ref. [3]),

$$\rho(\omega) = \frac{2|\omega/\Delta_a|^{2\gamma}}{2\pi\nu\Gamma(1+2\gamma)}$$

with $\alpha_{HLL} = 2\gamma$. The exponent $C = 1/2$ used in the manuscript for the helical edge states thus coincides with a regular (“spinless”) TLL, in contrast to $C = 1/4$ for spinful 1D liquids (see e.g. Ref.[10]). We have now included an abridged version of the above explanation in the manuscript, also including Ref. [3] in the list of references, where it used to read,

energy E_F [5, 12, 15, 16]. In a TLL, the tunnelling conductivity around such ZBA scales *universally*, that is, as a power law in both bias voltage and temperature, with the same scaling exponent α , related to the Luttinger parameter as $\alpha = C(K + K^{-1} - 2)$. Different from the spin-degenerate (“spinful”) parabolic dispersions of conventional 1D metallic systems ($C = 1/4$), quantum spin Hall (QSH) insulators host linearly dispersing 1D edge states, in which the spin polarity is locked to the crystal momentum (“helicity”) [6, 17]. TLL formation in such 1D helical systems ($C = 1/2$) [18] is not only of fun-

It now reads,

energy E_F [5, 12, 15, 16]. In a TLL, the tunnelling conductivity around such ZBA scales *universally*, that is, as a power law in both bias voltage and temperature, with the same scaling exponent α , related to the Luttinger parameter as $\alpha = C(K + K^{-1} - 2)$ [17]. Different from the spin-degenerate (“spinful”) parabolic dispersions of conventional 1D metallic systems ($C = 1/4$), quantum spin Hall (QSH) insulators host linearly dispersing 1D edge states, in which the spin polarity is locked to the crystal momentum (“helicity”) [6, 18]. For a helical Luttinger Liquid, strong spin orbit coupling causes the spin and chirality indices to coincide, and the bosonized Hamiltonian can be expressed as an effectively “spinless” Luttinger liquid with $C = 1/2$. TLL formation in such 1D helical

2. What is the definition of the strong coupling (the manuscript referred to $K < 1/2$), weak coupling (the manuscript referred to $K > 1/2$), and very strong interaction regime (the manuscript referred to $K < 1/4$)? How is the number chosen? It is confusing for general readers on the definition. Please explain it in a little more detail.

In general, many-body interactions are considered “strong” (“weak”) if Coulomb interactions between the particles and quasi-particles are stronger (weaker) than their kinetic energy. Regarding Eq. (1) of the main text, this naturally defines a threshold $K \approx 1/2$ where the Coulomb and kinetic energy terms are equal. We now clarify this in the manuscript, where it used to read,

many-body interactions. The Luttinger parameter can thus be expected to be highly tunable, ranging within its theoretical bounds, $0 < K < 1$, for repulsive (e.g. Coulombic) interactions, where $K = 1$ [$W(q = 0) = 0$] represents the limit of non-interacting electrons.

It now reads,

ing the many-body interactions. The Luttinger parameter can thus be expected to be highly tunable, ranging within its theoretical bounds, $0 < K < 1$, for repulsive (e.g. Coulombic) interactions, where $K = 1$ [$W(q = 0) = 0$] represents the limit of non-interacting electrons. A threshold $K \approx 1/2$ is naturally given where Coulomb and kinetic energy terms become equal, defining $K < 1/2$ as the limit of strong interactions.

Indeed, while most works seem agree that weak interactions are characterized by $K > 1/2$ [1, 2, 4-6], there seem to be some differences in definition what is considered the limit of strong interactions [5, 6]. The limit $K < 1/4$ [2, 4] bears particular significance as two-particle processes [2, 4] may lead to additional phenomena such as e.g. Wigner crystallization [6]. We are now further clarifying these considerations in the revised text, where it used to read:

vironments. Interestingly, despite being in the strong coupling limit ($K < 0.5$), where deviations from Eq. (1) may be expected due to strong interactions and a renormalised Fermi velocity, we show that the perturbative approximation can allow for qualitative predictions of its fundamental dependencies in terms of the Coulomb interaction strength and edge state dispersion. Eq. (1) can thus serve as an important indicator for the tunability of Coulomb correlations in a Tomonaga-Luttinger liquid.

It now reads:

vironments. Interestingly, despite being in the strong coupling limit ($K < 0.5$), where deviations from Eq. (1) may be expected due to strong interactions and a renormalised Fermi velocity, we show that the perturbative approximation can allow for qualitative predictions of its fundamental dependencies in terms of the Coulomb interaction strength and edge state dispersion. **In the limit of very strong interactions ($K < 1/4$) reached in some of our samples, two-particle back-scattering of helical electrons may become significant [4, 23], potentially leading to additional effects such as fractional Wigner crystallisation [23].** Eq. (1) can thus serve as an important indicator for the **strength and** tunability of Coulomb correlations in a Tomonaga-Luttinger liquid.

3. How does the Luttinger parameter K (or power-law) change with a distance to the edge? Fig. 1g only fits the K and zero-bias anomaly to those points at the author-defined edge, but not to those close to the “edge”, or slightly from the edge. I am wondering if the universal scaling is still valid for those areas and whether the universal scaling is still valid.

We thank the reviewer for this query, and for allowing us to clarify. Indeed, the electronic edge state can be seen to have a finite spatial extent (edge state width), with an overall enhancement of the LDOS close to the edge and exponential decay into the bulk. The superimposed charge density modulations close to the physical edge indeed make a precision definition of the edge location challenging which, we agree, should be taken into account.

If the electronic edge state is indeed a continuous 1D charge distribution, however, then the Luttinger parameter K should not exhibit any significant spatial variations depending where exactly it is measured across the edge. We confirm this in Figure.1 below, where we

show the spatial profile of K (panel c), extracted from fits to the universal scaling relation (Eq.(2), main text). In regions for which no data points are shown, our TLL fit did not converge owing to the presence of the QSH energy gap.

The additional data confirms that although K follows roughly the LDOS modulations seen in panel (d), variations in the precise value of K are minor, and within less than one standard deviation of the Y-edge histograms (compare Fig.3a, main text). For spectra with greater distance to the edge, the fitting error simply increases, due to the lower signal-to-noise ratio of the measured spectra and the weaker edge state signature overall (compare panel (g) of the revised Figure 1 below).

Our extraction of K is therefore robust with regard to the precise definition of the edge. This is further confirmed by fits to the different edge state LDOS maxima shown (revised Figure. 1, blue arrows) we find $K = (0.35 \pm 0.02)$ and $K = (0.37 \pm 0.02)$, respectively, in agreement with what is currently noted in the manuscript. None of our conclusions are therefore affected.

Figure 1: QSH gap and edge states. (a) STM height profile across the monolayer edge. (b) Measured dI/dV versus energy and distance X . (c) Luttinger parameter K extracted from TLL fits of the dI/dV in (b). (d) integrated LDOS as a function of distance X . The inset shows contrast between the dI/dV of the bulk and the edge, where the former shows a clear band gap of ~ 100 meV and the latter a V-shaped pseudogap near $E-E_F = 0$ meV.

In our revised Figure 1, we have now remove the vertical dashed line highlighting any particular edge position assumed. Also, we have removed any mention of a specific edge definition x_0 in main text or caption, and are now explicitly acknowledging the finite width of the edge state with LDOS modulations superimposed. Where it used to read,

a metallic edge state. We identify the precise position of the edge, electronically, from a profile of the energy-integrated LDOS (-20 to -40 meV) in Fig. 1f, which is seen to peak at the crystal edge and exponentially decay into the 2D bulk. From an exponential fit, we extract a decay length of (2.1 ± 0.2) nm, significantly larger compared to the crystal lattice constant ($a = 0.348$ nm), indicating that the observed LDOS enhancement is not due to localised atomic orbitals, but rather is an electronic effect due to the presence of metallic edge states.

It now reads,

a metallic edge state. We identify the precise position of the edge, electronically, from a profile of the energy-integrated LDOS (-20 to -40 meV) in Fig. 1f, which is seen to peak at which shows a finite width $\Delta x \approx 2$ nm with exponentially decay into the 2D bulk. From an exponential fit, we extract a decay length of (2.1 ± 0.2) nm, significantly larger compared to the crystal lattice constant ($a = 0.348$ nm). Spatial modulations, superimposed onto the overall edge state LDOS enhancement further indicate that the observed LDOS enhancement is not due to localised atomic orbitals, but rather are an electronic effect due to the presence of metallic edge state.

The final revised Figure 1 and caption are shown below:

FIG. 1. **Edge states in monolayer $1T'$ - WTe_2 .** **a**, STM topography of a monolayer $1T'$ - WTe_2 crystal (scale bar: 5 nm). Inset: Crystal structure of the $1T'$ - WTe_2 crystal structure with W atoms (blue) and Te atoms (orange). Blue arrows represent different edge terminations. **b,c**, Corresponding differential conductance (dI/dV) maps of the same island as in (a), measured in constant height mode. **d-f**, Spatial profile of the measured local density of state (LDOS) at 4.5 K, across the monolayer edge along the black arrow in (a), and compared to a corresponding STM height profile (d). We extract a gap of ~ 70 meV in the 2D bulk. The data in (f) has been obtained by integration of the LDOS in (e) between $E = -20$ meV to $E = -40$ meV and shows an overall enhancement near the edge with superimposed charge density modulations also observed in (b). The solid black line shows a fit to an exponential decay from which we extract a decay length of (2.1 ± 0.2) nm. Inset: Comparison of point spectra at bulk and edge, highlighting the bulk gap (black) and LDOS enhancement at the edge (blue) corresponding to the edge state. **g**, Evolution of the edge state LDOS from the edge into the bulk (see corresponding markers in (a)). The solid black lines are fits to TLL theory (see text). All spectra have been offset for clarity, with the zero position indicated (horizontal dashed lines).

4. The work presents the dielectric effect on the TLL and demonstrates a good fit for calculation, which is important to understanding the screening effect on the helical Luttinger liquids. However, there are only two data points of the ϵ_r . It is optional to get it better by adding one more datapoint if the authors have more data on other dielectrics, e.g., few-layer graphite or monolayer graphene. I fully understand the challenging in experiments, but this will make the work much higher standard in presentation and demonstration.

We thank the reviewer for acknowledging the challenging experiments. Indeed, these data are the result of months of measurements to obtain the necessary statistics for a precision extraction of the Luttinger parameter. Aside from being time-consuming, the experiments have also been very cost-intensive due to the excessive LHe usage at elevated temperatures, where the excessive LHe boil-off is significant. This is especially significant considering the current global LHe supply shortage.

Instead, in Figure 2 below we include two additional data points extracted from literature for two further dielectrics. One data point (Tao et al. 2022) has been obtained in our own lab from the additional analysis of published local probe spectroscopy data on epitaxial WTe_2 on the van-der-Waals metal $NbSe_2$ ($\epsilon_{NbSe_2} \rightarrow \infty$) [9]. Another data point (Fei et al. 2017) has been extracted from a published [8] transport experiment of h -BN encapsulated ($\epsilon_{hBN} \rightarrow 3.76$ [7]) WTe_2 . We note that both data points represent single measurements of K , rather than having been determined from statistical distributions, explaining the slightly larger error bars.

The second data bears particular significance, given that it was obtained from transport spectroscopy rather than from local probe spectroscopy. Accordingly, we have considered an additional parameter γ the universal scaling expression,

$$\frac{dI}{dV} = AT^\alpha \cosh\left(\gamma \frac{eV}{2k_B T}\right) \left| \Gamma\left(\frac{1+\alpha}{2} + \gamma \frac{ieV}{2\pi k_B T}\right) \right|^2$$

quantifying the symmetry of the two tunnel junctions to the TLL [10]. Different from the highly asymmetric tunnel barriers ($\gamma = 1$) assumed in local probe spectroscopy [1], in transport experiments, the two junctions are usually assumed approximately equal ($\gamma = 1/2$) [10], which we have explicitly considered for these fits. Consistent with Ref.[10], we further assume that tunnelling occurs into the bulk of the 1D liquid, rather than any end points (see Ref.[2] of the main text for further detail), allowing us to use the same expression as in point 1. above, relating the extracted power-law exponent α to the Luttinger parameter K .

Although, we do not have access to a full set of temperature-dependent raw-data for the transport spectroscopy experiment [8] making it challenging to unambiguously confirm universal scaling, the different modes of accessing the TLL's tunnelling density of states (transport vs. local-probe spectroscopy), different dielectrics, and different fit model, the additional data confirms well the overall scaling of K , in good agreement with our original data and model.

Figure 2: Tuning the Luttinger parameter via the dielectric constant. Grey diamonds, are Luttinger parameters for WTe_2 edges on two additional dielectrics (h -BN and $NbSe_2$), consistent with our data and calculations (shaded bands).

5. Since K is different for X and Y edges, a quick question is how the K will be at the intersection of the X and Y edges?

While X and Y edges are reasonably well-defined, they are also disordered. This becomes particularly apparent at corners (e.g. Fig.1a-c), which are rounded with bending radius over several nanometres. If we assume that in the vicinity of the edge the Fermi velocities approach approximately equal values, the primary difference between K_1 on the X-edge and K_2 on the Y-edge would be due to Coulomb interactions due to the differences in site resolved electron occupation at the conducting edges. The Luttinger parameter near the corner would then be approximately

$$(a_1 + a_2)/K^2 = a_1/K_1^2 + a_2/K_2^2$$

Where $0 < K_{1,2} < 1$ as per usual. We hope to investigate the spatial dependence at corners through further experiments and calculations in the future.

Other suggestions/comments that might be helpful:

1. In Ref. 10, the page number is wrong.

*We apologise for this mistake. The correct page number is Nature **397**, pages 598–601 (1999).*

2. I would suggest replacing the ref. 13 (PRL 86,143 (2001)) with PRL 77, 253 (1996). The leading author is the same. This is the first demonstration of the TLL in fractional quantum Hall edges.

We thank the reviewer for this suggestion. We have now changed Ref. [13] of the main text to the earlier report in Ref.[11].

Reviewer #2 (Remarks to the Author):

Jia and collaborators study the possibility of controlling the Luttinger parameter K in a one-dimensional (1D) helical Tomonaga Luttinger Liquid (TLL) that forms along the edges of the quantum spin Hall insulator $1T'$ -WTe₂. In a TLL, elementary excitations are replaced by collective modes. They consist of charge and spin density waves, with spin-charge separation, described as bosons rather than fermions. A single dimensionless parameter, the Luttinger parameter K , describes the correlation length of these collective modes. K depends on electrostatic interactions which are sensitive to the dielectric environment of the 1D TLL. Hence the idea to tune K by proximity to selected different substrates. To test this idea, Jia et al. study $1T'$ -WTe₂ monolayer crystals synthesised on highly-oriented pyrolytic graphite and on bilayer graphene islands by molecular-beam epitaxy.

The experimental challenge is the identification of the TLL in the first place. Jia et al. rely on the suppression of the local density of states (DOS) at the Fermi level in the tunneling spectra measured using a scanning tunneling microscope -a standard scheme. They find that this so called zero-bias anomaly (ZBA) shows universal scaling behavior as a function of bias voltage and temperature. The lineshape of the ZBA, its perfect scaling with temperature, and the changing scaling factor for different substrates are all consistent with the presence of a TLL. These experimental results are based on systematic temperature-dependent scanning tunnelling spectra, and on a detailed statistical analysis of tens of tunneling points along $1T'$ -WTe₂ edges. They show that K obeys normal distributions with distinct statistical mean for different substrates and crystal edges. These findings are corroborated both by numerical modelling to estimate the strength of the screened Coulomb interactions within the charge distribution along the edges, and by a material-specific real-space tight-binding model. The modelling further allows Jia et al to show that a Coulomb gap can neither account for the ZBA lineshape nor for its scaling.

Jia et al. propose a timely study with a very interesting result. The modelling appears reasonable, with a convincing fit of the ZBA and its scaling -although I am not fully proficient in the proposed theoretical modelling. On the other hand, these conclusions rely on experimental data that raise some questions and concerns I detail below. The authors need to address these issues before the work may be published.

We thank the referee for the positive assessment, highlighting that our study is “timely” and “very interesting”. We address the remaining queries and concerns below.

A. How do the authors identify the X and Y edges? There are very few, if any, long straight edges in the images presented in Fig.1 and Fig.S1. Reading the text, one expects to find well defined and long X and Y edges (the authors talk about “100–400 micrometer long edges for some crystals) when the quoted length in fact refers to the perimeter of a monolayer grain. It would be more appropriate and significant to refer to the typical length of relevant edges.

We apologize if our manuscript gave the impression that we have 100-400 micrometer long straight edges. This was not intended. MBE-growth of $1T'$ -WTe₂ is well-known [12] to yield comparatively small-diameter disordered islands with atomically corrugated edges [12]. In all our measurements, we have therefore focussed on sections of the edge which are reasonably straight locally, with comparatively low disorder levels, and similar to those seen in Fig.1 of the main text. The orientation of the edges can be determined from our STM images to be

either parallel (X) or perpendicular (Y) to the 1T'-WTe₂ atomic rows. We now further clarify this point in the manuscript, where it used to read,

of meV [23, 29], tunable by electric fields, local strain. A typical 1T'-WTe₂ monolayer crystal is shown in Fig. 1a, synthesised by van-der-Waals molecular-beam epitaxy (MBE) [23, 29] on highly-oriented pyrolytic graphite (HOPG) (see **Methods**), displaying well-defined edges terminated both along the Y- (perpendicular to the atomic rows) and X-direction (parallel to the atomic rows). The image shows part of a larger crystal, for which we achieve typical edge lengths (circumference) $L = 100 - 400$ nm.

It now reads,

of meV [25, 31], tunable by electric fields, local strain. A typical 1T'-WTe₂ monolayer crystal is shown in Fig. 1a, synthesised by van-der-Waals molecular-beam epitaxy (MBE) [25, 31] on highly-oriented pyrolytic graphite (HOPG) (see **Methods**). We achieve typical island circumferences $L = 100 - 400$ nm with disordered edges. Locally, however, we find straight and well-defined sections, with typical lengths between 10 – 20 nm that are terminated either perpendicular (Y-edge) or parallel (X-edge) to the atomic rows.

It is important to note, however, the relevant length setting the scale for the Luttinger parameter K is not the length of a particular sections of the disordered edge, but the length of the continuous 1D charge distribution in which the interacting electronic liquid is present. As we have also highlighted in our response to Reviewer #1, despite edge roughness and disorder giving rise to superimposed charge density modulations, the LDOS shows an overall enhancement at the crystal edge (see e.g. Fig.1f and similar data in Fig. 3 below), and is continuous along the islands' circumference (Fig.S1). Indeed, as shown in Fig.1b-c, we find that the LDOS enhancement – though modulated – is continuous even at adjoining corners between different edge types (Fig.1b-c, main text). The relevant length-scale determining the electrostatic energy of the screened Coulomb interactions within the edge state's 1D charge distribution is therefore indeed be the island circumference ($L = 100 - 400$ nm). We now clarify this point further in the revised manuscript, where it used to read,

insulating 2D bulk [23, 31]. A larger area of the same crystal can be viewed in Section S1 of the Supplementary Information. We note that we do observe semi-metallic

It now reads:

insulating 2D bulk [25, 33]. A larger area of the same crystal can be viewed in Supplementary Fig. 1, showing that the electronic edge state is largely continuous along the island's circumference, even in the presence of local edge roughness and disorder, and regardless of edge direction or termination, and at adjoining corners.

The relevant length-scale determining the electrostatic energy of the screened Coulomb interactions within the edge state's 1D charge distribution is therefore set by the islands' circumference. Superimposed spatial modulations in the charge density observed can be shown to arise from edge roughness even in otherwise perfect monolayer crystals (Supplementary Fig. 7). We note that

B. It is often quite challenging to precisely identify the position of a step edge in topographic STM images. I am not convinced that Jia et al. have properly identified the edge in Figs.1d, 1e and 1f. The Z-profile in Fig.1d actually increases beyond the point identified as the edge, suggesting the monolayer edge might actually be up to 1nm further to the right. The proper identification will have an impact on the assignment of spectral features to an edge state, and on the fitting of the decay length in Fig.1f. Why would the STM tip height increase beyond the edge of the 1T'-WTe₂ flake in a constant current scan?

We thank the reviewer for acknowledging the challenge in identifying electronic edge state positions in STM. As the reviewer is aware, "topographic" images contain both z-height and electronic information combined, as the tunnel current is an integral over the sample's density of states within the bias window. The topography image from which the original height profile in Fig.1d had been extracted was recorded at +1V, far greater than the spectral range of features seen in Fig.1e, and making it difficult to correlate with the low-energy features in the spectroscopy. Indeed, any higher-energy feature in the LDOS may easily account for the small local variations (a few percent) in the apparent z-height as seen in Fig.1d.

However, to further confirm the step edge position, we reproduce Fig.1d-f of the main text below, but with the inclusion of an additional STM z-height profile, that was independently obtained during spectroscopy line-mapping of the data set in (b) and recorded at a smaller bias of +0.2V. Comparing both height-profiles confirms the correct location of the step edge ($x \approx 9\text{nm}$). "Topographic" features near the edge indeed in the energy range indeed coincide well with the spectroscopy features seen in (b) and (d) below.

In alignment with our response to Reviewer #1, we further acknowledge that the finite width of the electronic edge state's LDOS enhancement near the step edge ($\Delta x \approx 2$ nm), with exponential decay into the bulk, and superimposed charge density modulations, needs to be understood as part of its electronic signature. Where it used to read,

a metallic edge state. We identify the precise position of the edge, electronically, from a profile of the energy-integrated LDOS (-20 to -40 meV) in Fig. 1f, which is seen to peak at the crystal edge and exponentially decay into the 2D bulk. From an exponential fit, we extract a decay length of (2.1 ± 0.2) nm, significantly larger compared to the crystal lattice constant ($a = 0.348$ nm), indicating that the observed LDOS enhancement is not due to localised atomic orbitals, but rather is an electronic effect due to the presence of metallic edge states.

It now reads,

a metallic edge state. We identify the precise position of the edge, electronically, from a profile of the energy-integrated LDOS (-20 to -40 meV) in Fig. 1f, which is seen to peak at which shows a finite width $\Delta x \approx 2$ nm with exponentially decay into the 2D bulk. From an exponential fit, we extract a decay length of (2.1 ± 0.2) nm, significantly larger compared to the crystal lattice constant ($a = 0.348$ nm). Spatial modulations, superimposed onto the overall edge state LDOS enhancement further indicate that the observed LDOS enhancement is not due to localised atomic orbitals, but rather are an electronic effect due to the presence of metallic edge state.

Crucially (also referring to our response to Reviewer #1), we would like to re-emphasize, that there are no significant variation in the Luttinger parameter K across the edge state, despite the presence of charge density modulations. This further confirms the continuity of the edge charge distribution and that our analysis is robust with regard to the definition of the edge location and its spectral features. Our original conclusions remain unaffected.

C. The authors claim that the conductance maps of the monolayer 1T'-WTe2 flake in Figs.1b and 1c reflect "the expected QSH electronic signature of an enhanced local density of states (LDOS) at the crystal edge around an insulating 2D bulk". Looking at Fig.1 and Fig.S1, I find only one feature clearly compatible with a 1D edge state, namely along the lower right Y edge. The other edges, especially the X edges, show either no enhanced local DOS or elongated 1D-like features extending perpendicular rather than parallel to the edge. Moreover, the brighter 1D spectral feature along the lower right Y edge does not seem to sit exactly on the 1T'-WTe2 monolayer edge (see comment B). Why would there only be one edge with a parallel 1D DOS feature, and why do the authors identify features along the Y edges extending perpendicular into the 2D bulk as an edge state?

We need to respectfully disagree with this observation. Fig.1b-c clearly shows (i) an insulating 2D bulk (with the density of states matching that of the surrounding substrate) and (ii) a finite (enhanced) density of states at the edges all-around the circumference of the island (Fig.S1), independent of edge type or local disorder.

This is a real material, with realistic disorder, and with LDOS probed at the atomic-level. In any condensed matter system, disorder can be expected to lead to some non-uniformity of the local electronic structure – even in a topological insulator. Similar edge state mapping, published by Jia et al. [13] on WTe_2 is reproduced in the figure below. Although the spatial resolution of this data is somewhat limited compared to ours and the colour-scale partly saturated near the edge, the presence of similar disorder-induced non-uniformity in the edge state and anisotropy of LDOS features can be ascertained, including also the elongated 1D-like modulations parallel to the Y-edges and perpendicular to the X-edges as pointed out by the reviewer. Despite disorder-induced charge density modulations, an overall enhancement at the edge was reported [13], consistent with our own observations (e.g. shown for the Y-edge in Fig.1f of the main text, and for the X-edges in Figure 3 below.

Figure 3: Continuity and Charge density fluctuations of the WTe_2 X-edge. (a) dI/dV map of the crystal in Figs. 1 and S1 of the manuscript. The arrows indicate the line cuts in (b), across the X-edge, showing an overall LDOS enhancement close to the edge with exponential decay into the bulk, and with comparable decay lengths ($\xi \approx 2\text{nm}$) (Figures 3c-d), consistent with that of the Y-edge (Fig.1f, main text) ($\xi \approx 2.1 \pm 0.2\text{nm}$). This strongly points at the same origin of the X and Y edge being topological edge states despite the difference in the appearance of their LDOS modulations.

While a detailed investigation of the nature of observed modulations goes beyond the scope of the present paper, below we show that even in the absence of 1D correlations [6], realistic edge roughness can indeed lead to charge density modulations at the edge – even in an otherwise ideal QSH system.

To confirm this, we have performed real space tight binding calculations of the edge state (Figure 4 below) using realistic band parameters [9] to quantitatively reproduce the low energy features in the LDOS of disordered WTe_2 edges. Self-consistently calculating the electron densities and local density of states near the edge for realistic disorder/roughness as extracted from atomic-resolution STM images (panels a and e) shows that the LDOS maps

(panels f-h), can indeed reproduce the observed LDOS modulations well, including even the “elongated 1D like features” perpendicular to the X-edge, highlighted by the reviewer.

Importantly, only complete unit cells have been included in our representation of edge roughness, ensuring that dangling bonds or other other effects cannot not give rise to the LDOS enhancement observed, and that the LDOS modulations indeed are a consequence of realistic edge disorder, despite the topology enforced by our 4-band WTe_2 model. We have now included the below data, alongside a short description, in the revised supplementary information.

Figure 4 Real-space tight binding calculation of disordered QSH edges. (a-d) STM image of the X-edge of the sample in Figs. 1 and S1, and corresponding dI/dV maps. Atomic-level edge roughness, extracted from the STM image in a and used for the real-space TB calculations (f-h).

D. The statement that “An enhanced LDOS persists along the sample edge for all energies, but the bulk is insulating between approximately -60 meV and +50 meV, with the LDOS matching that of the HOPG substrate” in the legend of Fig.S1 is not compatible with the data it is supposed to describe. The bulk DOS is low and comparable to that of the HOPG substrate only between -40mV and +20mV bias. Above +20mV, it is larger than the DOS of HOPG, and even larger than the step edge DOS above +40mV.

We apologize for this inaccuracy and have updated the numbers accordingly.

E. The legend for panel g in Fig.1 is confusing. It says “Evolution of of (typo) the edge state LDOS from the edge into the bulk”. But the five dark blue spectra are all measured at a position identified by the authors as the step edge. Are they measured at the very same position or at different locations along the step edge? I suggest to really show the evolution of the spectra away from the step edge and show the reproducibility of the edge spectra in a separate panel.

All blue traces of Fig.1g were recorded at different locations at the edge, separated by 0.1nm. We have now updated Figure 1 for larger increments (reproduced below), illustrating the evolution of the edge state more clearly. We have also fixed the above-mentioned typo.

Reviewer #3 (Remarks to the Author):

This manuscript presents STM study of the topological edge state in the quantum spin Hall insulator 1T'-WTe₂. The authors claim helical Luttinger liquid physics in these 1D channels by the observation of bias- and temperature-dependent tunneling spectra that show power-law behavior. While I find this topic interesting, I have some concerns that need to be clarified before I could recommend publication of the manuscript.

We thank the reviewer for his interest in the topic and for giving us the opportunity to address his queries and suggestions.

The topological edge state (TES) presented in this manuscript does not look to be a continuous 1D channel (Fig. 1 and Fig. S1). For example, in Fig. S1 it looks disconnected with lots of scatterers and breaks.

We like to re-emphasize that this material is a realistic condensed matter system, investigated at the atomic-level. As argued in our response to Reviewer #2, despite the presence of inhomogeneity, the edge state exhibits an otherwise continuous LDOS enhancement (compare Figure 3 and 4 above) along the islands' circumference (see also Fig.S1), but with LDOS modulations superimposed. Both LDOS enhancement and superimposed spatial modulations are in good agreement with published work [13]. Non-uniformity in the charge density can be shown to arise from disorder – even in a perfect topological insulator without scattering – as our real-space tight-binding model confirms (Fig. 4 above).

Besides, the X-edge and the Y-edge show distinct Fermi velocities and completely different band shapes (Fig. 3d-e). Based on the above reasons the effective chain length of the 1D channels should be around 10 ~ 20 nm as can be seen in Fig. S1. This is because theory predicted that the impurity potential in a 1D TLL with repulsive interaction would renormalize to infinity in the low-energy limit, thus cutting a 1D TLL into shorter channels (e.g., chapter 27 in the book “Bosonization and Strongly Correlated Systems” by Alexander O. Gogolin et al.). For a $L \sim 20$ nm channel with a Fermi velocity $v_F \sim 1.16 \times 10^5$ m/s (this value is from the manuscript), the quantum confinement energy is about $E_0 = \hbar v_F \pi / L \sim 12$ meV, well above the experimental energy resolution. This should result in a hard gap at the Fermi level (e.g., Ref. 16 and arXiv: 2108.03829), rather than the V-shaped power-law gap observed in this manuscript. The authors need to carefully clarify this issue.

We agree with the reviewer that 1D edges as short as 10-20nm would give rise to a finite-length TLL, with fundamentally different spectroscopic signature – a hard-gap over several meV. Such signature has not been observed in any of our experiments and thus eliminates this possibility. In all our measurements, we have observed a clear V-shaped signature at the edge, consistent with tunnelling into a continuous TLL – regardless of substrate type or edge – that exhibits universal scaling and dielectric tunability in agreement with our theoretical models.

As for the reasons why we do not observe any stronger “segmentation” of the edge, we would like to re-emphasize the 1D helical nature of electronic states in this QSH system. Single-particle back-scattering off non-magnetic impurity potentials is forbidden by time-reversal symmetry. Effects of e.g. Anderson localization due to disorder should therefore be weak and size quantisation, if any, would be expected to arise only from the finite sample circumference. While disorder-induced spatial modulations are observed along the edge, disorder effects indeed seem to be too weak to cause any “hard-wall” (TLL-in-a-box) confinement potential strong enough to cause segmentation of the edge state. As expected from these

considerations, the TLL signature is indeed consistent with an infinite-length TLL, given the continuous 1D edge charge distribution.

In alignment with our response to Reviewer #2, we now clarify this issue in the revised text, where it used to read,

of meV [23, 29], tunable by electric fields, local strain. A typical 1T'-WTe₂ monolayer crystal is shown in Fig. 1a, synthesised by van-der-Waals molecular-beam epitaxy (MBE) [23, 29] on highly-oriented pyrolytic graphite (HOPG) (see **Methods**), displaying well-defined edges terminated both along the *Y*- (perpendicular to the atomic rows) and *X*-direction (parallel to the atomic rows). The image shows part of a larger crystal, for which we achieve typical edge lengths (circumference) $L = 100 - 400$ nm.

It now reads,

of meV [25, 31], tunable by electric fields, local strain. A typical 1T'-WTe₂ monolayer crystal is shown in Fig. 1a, synthesised by van-der-Waals molecular-beam epitaxy (MBE) [25, 31] on highly-oriented pyrolytic graphite (HOPG) (see **Methods**). We achieve typical island circumferences $L = 100 - 400$ nm with disordered edges. Locally, however, we find straight and well-defined sections, with typical lengths between 10 - 20 nm that are terminated either perpendicular (*Y*-edge) or parallel (*X*-edge) to the atomic rows.

And where where it used to read,

insulating 2D bulk [23, 31]. A larger area of the same crystal can be viewed in Section S1 of the Supplementary Information. We note that we do observe semi-metallic

It now reads

insulating 2D bulk [25, 33]. A larger area of the same crystal can be viewed in Supplementary Fig. 1, showing that the electronic edge state is largely continuous along the island's circumference, even in the presence of local edge roughness and disorder, and regardless of edge direction or termination, and at adjoining corners. The relevant length-scale determining the electrostatic energy of the screened Coulomb interactions within the edge state's 1D charge distribution is therefore set by the islands' circumference. Superimposed spatial modulations in the charge density observed can be shown to arise from edge roughness even in otherwise perfect monolayer crystals (Supplementary Fig. 7). We note that

*Regarding the role of the Fermi velocities and band shapes, strong scattering would only be expected at atomically sharp corners between *X* and *Y* edges. However, as shown in Fig.1 of the main text, even for well-defined and straight *X* and *Y* edges, the adjoining corners are rounded with a bending radius of a several nanometres. In the vicinity of corners, the band shapes may not be well-defined, thus not posing a significant source of scattering as we also argued in our response to Reviewer #1.*

The authors claim that the TLL physics applies to the entire QSH gap range due to the linear dispersion of the TES. However, as can be seen in Fig. 2a-b, I do not see well power-law fits over the entire gap energy range, but instead fits over a smaller range (-20 meV to 15 meV) are shown.

The main data set of Fig.1g (main text) indeed shows fits over the entire energy range of the QSH energy gap (-40meV to +20meV) with rather good fit quality. This indicates that the edge state's dispersion is linear to good approximation. However, we also note that in this particular crystal, the Fermi energy $E - E_F = 0$ is centred nearly mid-gap indicating negligible doping. We do observe slight variation in gap size and doping level from crystal to crystal. Reduction in fit quality is sometimes observed close to the 2D bulk band edges, indicated e.g. by grey shading in Figs.2a,b. Here, the 2D DOS of the band tails can contribute to the total LDOS measured – especially at higher temperature – and leads to reduced fit quality.

However, despite these effects, in all our data sets we maintain a typical fitting range of 20-30 meV around E_F . This compared to an average QSH gap size of 55 meV [12] still constitutes a significant percentage of the total gap. We have clarified the fit range in the main text, where it used to read

Particular to 1D helical systems, the linear (Dirac) dispersion supports the bosonisation picture of TLL theory over a much larger energy range. Contrasting the spinful parabolic dispersions of conventional metals, which deviate from a linear approximation already at moderately low energy, the 1T'-WTe₂ edge state dispersion remains linear over nearly the entire QSH band gap [32] (compare Fig. 3d-e), consistent with the range of our TLL fits. From 5 edge spectra in Fig. 1g, we extract

It now reads:

Particular to 1D helical systems, the linear (Dirac) dispersion supports the bosonisation picture of TLL theory over a much larger energy range. Contrasting the spinful parabolic dispersions of conventional metals, which deviate from a linear approximation already at moderately low energy, the 1T'-WTe₂ edge state dispersion remains linear over **a wide energy range within the QSH band gap** ~~nearly the entire QSH band gap~~ [33] (compare Fig. 3d-e), consistent with the range of our TLL fits (**typically -20 meV to +15 meV**). From 5 edge spectra in Fig. 1g,

The experimental data also deviates from the power-law fit curves even within such a smaller energy range (e.g., 25 K in Fig. 2a and 10 K in Fig. 2b). In addition, the shaded area representing statistical confidence shown in Fig. 2c is quite broad $\sim 50\%$ (width is about 4 a.u. while the data point is about 8 a.u. at -30 eV/kBT). To support the authors' claim, the authors need to present better data or to refine their data analysis.

As noted in the point above, the contribution of band tails to the LDOS in the vicinity of the bulk band edges can lead to reduction in fit quality, especially at higher temperatures above 4K. However, despite such effects we agree that the shaded confidence bands in Figs.2c,d (0.45 ± 0.15) for HOPG and (0.98 ± 0.15) for BLG) represent an overestimation of the error. The widths of the bands were originally chosen to reflect the standard deviation of the statistical distributions in Figs.3a-b. However, given that each data set in Figs.2c,d represent

a single measurement of K at a specific location (not the measurement of an ensemble over different locations), it can indeed not be justified to consider the standard deviation for the error. For a single measurement of K , a better estimation is the mean absolute error across the individual fits at different measurement temperature, for which we find (0.45 ± 0.03) for HOPG and (0.98 ± 0.03) for BLG. The revised analysis is shown in Figure 7c,d below, in which we further demonstrate universal scaling over a wider energy range.

Figure 5: Revised Fig.2 of the main text with improved data representation of universal scaling in panels b and c.

To clarify the error estimate, we have amended the main text, where it used to read,

parameter $eV/k_B T$. Following this procedure, all spectra collapse onto a single universal curve (Figs. 2c,d) with power-law exponents $\alpha = 0.45 \pm 0.15$ ($K = 0.40 \pm 0.05$) for WTe_2/HOPG and $\alpha = 0.98 \pm 0.15$ ($K = 0.27 \pm 0.03$) for WTe_2/BLG . The uncertainty in the fitted values of α are indicated by the shaded confidence bands in the figure, and agree in magnitude with the standard deviations determined in the statistical analysis of Fig. 3. As

It now reads,

parameter $eV/k_B T$. Following this procedure, all spectra collapse onto a single universal curve (Figs. 2c,d) with power-law exponents $\alpha = 0.45 \pm 0.03$ ($K = 0.40 \pm 0.01$) for WTe_2/HOPG and $\alpha = 0.98 \pm 0.03$ ($K = 0.27 \pm 0.01$) for WTe_2/BLG . The uncertainty in the fitted values of α , indicated by the shaded confidence bands in the figure, arises from the mean absolute error of TLL fits across the different measurement temperatures. As a further

REFERENCES

1. Stühler, R., et al., *Tomonaga–Luttinger liquid in the edge channels of a quantum spin Hall insulator*. Nature Physics, 2020. **16**(1): p. 47-51.
2. Li, T., et al., *Observation of a Helical Luttinger Liquid in InAs/GaSb Quantum Spin Hall Edges*. Physical Review Letters, 2015. **115**(13): p. 136804.
3. Braunecker, B., C. Bena, and P. Simon, *Spectral properties of Luttinger liquids: A comparative analysis of regular, helical, and spiral Luttinger liquids*. Physical Review B, 2012. **85**(3): p. 035136.
4. Maciejko, J., et al., *Kondo Effect in the Helical Edge Liquid of the Quantum Spin Hall State*. Physical Review Letters, 2009. **102**(25): p. 256803.
5. Ziani, N.T. , et al., *Charge and spin density in the helical Luttinger liquid*. EPL (Europhysics Letters), 2016. **113**(3): p. 37002.
6. Ziani, N.T., F. Crépin, and B. Trauzettel, *Fractional Wigner Crystal in the Helical Luttinger Liquid*. Physical Review Letters, 2015. **115**(20): p. 206402.
7. Laturia, A., M.L. Van de Put, and W.G. Vandenberghe, *Dielectric properties of hexagonal boron nitride and transition metal dichalcogenides: from monolayer to bulk*. npj 2D Materials and Applications, 2018. **2**(1): p. 6.
8. Fei, Z., et al., *Edge conduction in monolayer WTe_2* . Nature Physics, 2017. **13**(7): p. 677-682.
9. Tao, W., et al., *Multiband superconductivity in strongly hybridized $1'-WTe_2/NbSe_2$ heterostructures*. Physical Review B, 2022. **105**(9): p. 094512.
10. Bockrath, M., et al., *Luttinger-liquid behaviour in carbon nanotubes*. Nature, 1999. **397**(6720): p. 598-601.
11. Chang, A.M., L.N. Pfeiffer, and K.W. West, *Observation of Chiral Luttinger Behavior in Electron Tunneling into Fractional Quantum Hall Edges*. Physical Review Letters, 1996. **77**(12): p. 2538-2541.
12. Tang, S., et al., *Quantum spin Hall state in monolayer $1T'-WTe_2$* . Nature Physics, 2017. **13**(7): p. 683-687.
13. Jia, Z.-Y., et al., *Direct visualization of a two-dimensional topological insulator in the single-layer $1T'-WTe_2$* . Physical Review B, 2017. **96**(4): p. 041108.

REVIEWER COMMENTS

Reviewer #1 (Remarks to the Author):

I have read through the response and manuscript. All the questions raised by the reviewers have been well addressed. I believe the manuscript is now in good shape to be published, considered a minor comment: The revised Fig.1c missed some data near $X=6\text{nm}$. The authors are expected to address why they are missing.

Reviewer #3 (Remarks to the Author):

I thank the authors for their detailed explanations which have clarified almost all of my concerns. In particular, I like the argument that backscattering is avoided in QSH edge states to ensure the “connectedness” of the 1D electronic structure.

However, I have one remaining question as I mentioned in the last round: I am really curious whether the 1D topological edge state as a whole can be viewed as a single well-defined TLL, given the fact that the X and Y edges look so different in their electronic structure, and the fact that the observed K values of the X and Y edges are different. Could it be possible that the topological edge state is continuous but the TLL-related electronic feature is discontinuous (in other words, X and Y edges host distinct TLLs)? In this regard, a detailed spatial dependence of the experimental K value from X to Y edges along the edge direction would help to clarify this issue (which was mentioned by referee 1 in the last round). This issue is important because it determines the actual channel length of the TLL.

Detailed Responses to the Reviewers

We thank the reviewers for their comments. Below we address remaining queries.

Reviewer #1 (Remarks to the Author):

I have read through the response and manuscript. All the questions raised by the reviewers have been well addressed. I believe the manuscript is now in good shape to be published, considered a minor comment: The revised Fig.1c missed some data near $X=6\text{nm}$. The authors are expected to address why they are missing.

We thank the reviewer for the positive assessment and for acknowledging our clarifications. For the remaining query, we believe that Reviewer #1 is referring to panel c of Figure 1 in our previous response letter (reproduced below), which shows the Luttinger parameter K as extracted for points across the monolayer edge from universal scaling fits. In the range $X = 5.2\text{ nm}$ and $X = 6.2\text{ nm}$, no data points are shown, as here fits to Tomonaga-Luttinger liquid (TLL) theory did not converge. This is likely due to a low signal-to-noise level of the edge state LDOS measured in this range, better seen in Figure 2 below (grey trace at $X = 6\text{nm}$). The spectrum shows the TLL feature at $E=E_F$, but at very low intensity within the noise, allowing the QSH bulk gap to dominate. We had noted the non-convergence of the TLL fit within this range in the context of point 3. of our previous response letter.

For values at around $X=5.2\text{nm}$, the edge state LDOS slightly increases owing to charge density modulations of the edge state, also seen in the integrated LDOS of Figure 1d below. This had allowed us to obtain two more data points in this range before the edge state signal again vanishes within the noise.

Figure 1: QSH gap and edge states. (a) STM height profile across the monolayer edge. (b) Measured dI/dV versus energy and distance X . (c) Luttinger parameter K extracted from TLL fits of the dI/dV in (b). (d) integrated LDOS as a function of distance X . The inset shows contrast between the dI/dV of the bulk and the edge, where the former shows a clear band gap of ~ 80 meV and the latter a V-shaped pseudogap near $E-E_F = 0$ meV.

Figure 2: Comparison between the electronic states of the bulk (black line), of a representative point between the edge and bulk (grey line), of the edge (blue line), and a TLL fit of the edge spectrum (red line). This comparison shows that the electronic states from regions between the edge and the bulk (such as the state at $X = 6$ nm) may behave as a quantum spin Hall insulator, as in the bulk.

Informal Comments of Reviewer #2:

What I can say about the revised manuscript is that I still have very serious doubts about the edge state analysis along the entire circumference of the flake. The main argument of the authors to counter the referee's questions on this issue is that the sample is a real system with defects, and that other published edge states are not better defined than theirs (Jia et al Phys. Rev. B 96, 041108(R) 2017). This does not sound convincing at all. In particular for a feature that ought to be robust against defects and taking a paper as example in which I think the edge states identification has similarly weak and questionable as in their analysis.

Amongst all quantum spin Hall candidates to date (Refs. [1, 2] for extensive reviews), WTe_2 is one of the most extensively studied examples [3-10]. Demonstrations of edge states include ballistic quantum spin Hall edge transport [3, 4], quantized and up to 100K [4], ARPES measurements [5], local probe spectroscopy [5, 7], and real-space visualization / mapping [7, 8, 10] of edge states. While our work goes significantly beyond prior work, as it presents the first systematic investigation of screened 1D Coulomb interactions and the formation of a Luttinger Liquid, it does build on previous papers in terms of established edge state identification and analysis.

Indeed, the spectroscopic signatures in our work are perfectly consistent with all prior reports of 1D edge states in WTe_2 by scanning tunnelling microscopy (see in particular Refs. [5, 7, 9, 10]). We would argue that labelling our edge state identification and analysis as "weak and questionable", therefore not only discards an isolated paper or two, but instead ignores a significant body of published literature [3-10]. We cannot see how this could possibly be justified.

We further like to re-emphasize very strongly that the TLL signatures resolved are simply inconsistent with any signature expected from a finite-length TLL. The presence of the latter would be obvious through a hard gap of tens of meV bounded by coherence peaks (see e.g. Ref. [11]). Instead, we observe experimental signatures of a TLL in the infinite limit that is consistent with universal (power-law) scaling of the LDOS. This implies that even if finite size effects were present, the quantization energy would be below the resolution limit set by thermal broadening ($3.5k_B T = 1.4$ meV at 4.5K) and any measurable signatures from a finite length TLL can be safely excluded.

Further evidence pertaining the continuity of the edge state LDOS, and the TLL feature, even in the vicinity of corners, is given by the additional data provided in our reply to Reviewer #3 below.

Reviewer #3 (Remarks to the Author):

I thank the authors for their detailed explanations which have clarified almost all of my concerns. In particular, I like the argument that backscattering is avoided in QSH edge states to ensure the “connectedness” of the 1D electronic structure.

We thank the reviewer for acknowledging our clarifications, in particular pertaining the continuity of the 1D helical edge state.

However, I have one remaining question as I mentioned in the last round: I am really curious whether the 1D topological edge state as a whole can be viewed as a single well-defined TLL, given the fact that the X and Y edges look so different in their electronic structure, and the fact that the observed K values of the X and Y edges are different. Could it be possible that the topological edge state is continuous but the TLL-related electronic feature is discontinuous (in other words, X and Y edges host distinct TLLs)? In this regard, a detailed spatial dependence of the experimental K value from X to Y edges along the edge direction would help to clarify this issue (which was mentioned by referee 1 in the last round). This issue is important because it determines the actual channel length of the TLL.

We thank the reviewer for raising this point. It is indeed an interesting question whether and how the two distinct TLLs on X- and Y-edges are adjoined over the corners of the WTe_2 crystal. In our previous letter, also in response to Reviewer #1, we had noted that we expect a smooth transition from one K parameter to another, given that the corners are not atomically sharp, but have an appreciable bending radius of a few nanometres, and that hence the Fermi velocity could be expected to be an interpolation of X- and Y-edge values.

To further support this point, we have extracted spatially averaged point spectra from the LDOS maps shown in Fig.1b,c and Fig.S1. While the energy resolution in this data is limited because of the spatial mapping, the transition from X- to Y-edge across the corner can be clearly observed from the changes in the power-law scaling parameter α . At the corner, the edge state LDOS remains finite, but the ZBA is slightly less well-defined and a bit asymmetric. Fits to TLL theory (black lines) suggest an average K which lies between those of the X- and Y-edge values.

Our detailed spatial dependence therefore suggests that along the edge, and even around corners, the TLL-related feature remains continuous and reasonably well-defined, and that K can be represented by an interpolation of the properties of the two distinct TLLs.

Figure 3: Evolution of the Luttinger parameter across corners. *a*, STM topography image of the same island as shown in Fig.1 of the main text. *b*, Spatially averaged point spectra at the locations highlighted by coloured markers in *a*. Black lines are fits to TLL theory. *c*, Extracted Luttinger parameter K .

Other changes to the manuscript / SI:

We note that, although this was not requested, we have further improved on our data presentation of the universal scaling (previously requested by Reviewer #3) by including double-logarithmic plots of the data (Supplementary Fig. S2 in the supplementary information). All previously requested panels in main text and SI remain the same.

- [1] Cao, C. and J.-H. Chen, *Quantum Spin Hall Materials*. *Advanced Quantum Technologies* **2**, 10 (2019).
- [2] Lodge et al., *Atomically Thin Quantum Spin Hall Insulators*. *Advanced Materials* **33**, 22 (2021).
- [3] Fei et al., *Edge conduction in monolayer WTe_2* . *Nature Physics* **13**, 7 (2017).
- [4] Wu et al., *Observation of the quantum spin Hall effect up to 100 kelvin in a monolayer crystal*. *Science*, **359**, 6371 (2018).
- [5] Tang et al., *Quantum spin Hall state in monolayer $1T'-WTe_2$* . *Nature Physics* **13**, 7 (2017).
- [6] Song et al., *Observation of Coulomb gap in the quantum spin Hall candidate single-layer $1T'-WTe_2$* . *Nature Communications* **9**, 1 (2018).
- [7] Jia et al., *Direct visualization of a two-dimensional topological insulator in the single-layer $1T'-WTe_2$* , *Physical Review B* **96**, 4 (2017).
- [8] Shi et al., *Imaging quantum spin Hall edges in monolayer WTe_2* . *Science Advances* **5**, 2 (2019).
- [9] Zhao et al., *Strain Tunable Semimetal-Topological-Insulator Transition in Monolayer $1T'-WTe_2$* . *Physical Review Letters* **125**, 4 (2020).
- [10] Maximenko et al., *Nanoscale studies of electric field effects on monolayer $1T'-WTe_2$* . *npj Quantum Materials* **7**, 1 (2022).
- [11] Jolie et al., *Tomonaga-Luttinger Liquid in a Box: Electrons Confined within MoS_2 Mirror-Twin Boundaries*. *Physical Review X* **9**, 1 (2019).

REVIEWER COMMENTS

Reviewer #2 (Remarks to the Author):

I have read the latest referee reports and the corresponding responses of the authors with attention. While I do appreciate the efforts invested in this work, I remain skeptical about the assignment of the features observed along the X edges to edge states.

Below, I outline the spatial extent of the enhanced LDOS along the X and Y edges, which Jia and collaborators associate with edge states. As already mentioned in my first report, I agree that the feature developing along the Y edge seems consistent with an edge state. However, I find it questionable to associate the bright features at the X edge, which extend perpendicular rather than along the edge as I have highlighted in white in the conductance maps below, with topologically protected edge states. This is a major difference for two features whose physical origin is supposed to be the same. I emphasize that my concern is not the discontinuities along the X edges (which can indeed be explained in terms of structural disorder of the edge as the authors show in their revised supplementary material), but rather on this distinct topology (parallel vs perpendicular to the edge).

To demonstrate that the features along the X edges are indeed edge states, Jia et al., in their first rebuttal, show $dI/dV(V)$ traces extracted from a conductance map at -70meV . Two comments about these figures reproduced below: while the distance from the edge mentioned in panel b may be correct near the black dotted line, they are not correct for locations near and to the left of the red dotted line which are clearly further away from the edge. Moreover, to find a similar exponential decay (within 26%) of the LDOS with distance from the edge, the authors need to shift the edge position by about 1,2 nm (vertical black lines in panels c and d below) between the two sites located 2,5 nm from each other. These all point at an ill-defined nature of these enhanced features near the step edge.

In summary, the manuscript is rather convincing with respect to the impact of the substrate's dielectric properties on the Y edges, but not with respect to the discussion of the X edges. For this reason, I do not support publication of this manuscript in its present form.

Reviewer #3 (Remarks to the Author):

The authors have performed additional data analysis in response to my question in the last round. The additional experimental curves in the transition region between the X and Y edges, however, look not very well fit to TLL behavior, as also mentioned by the authors. Therefore, I am not fully convinced by their interpretation. At this stage, it might be unfair to stop the manuscript from publication, since there were similar experiments done on other QSH material systems. The experimental findings here are likely to trigger further investigations into this field. I would in principle agree on the publication of the manuscript, although I disagree that the difference in the behavior of X and Y edges is sufficiently investigated and explained.

Detailed Responses to the Reviewers

We again thank all reviewers for the time spent to review the manuscript and our revisions, which believe further strengthened the paper.

Response to Reviewer #2:

I have read the latest referee reports and the corresponding responses of the authors with attention. While I do appreciate the efforts invested in this work, I remain skeptical about the assignment of the features observed along the X edges to edge states. Below, I outline the spatial extent of the enhanced LDOS along the X and Y edges, which Jia and collaborators associate with edge states. As already mentioned in my first report, I agree that the feature developing along the Y edge seems consistent with an edge state. However, I find it questionable to associate the bright features at the X edge, which extend perpendicular rather than along the edge as I have highlighted in white in the conductance maps below, with topologically protected edge states. This is a major difference for two features whose physical origin is supposed to be the same. I emphasize that my concern is not the discontinuities along the X edges (which can indeed be explained in terms of structural disorder of the edge as the authors show in their revised supplementary material), but rather on this distinct topology (parallel vs perpendicular to the edge).

To demonstrate that the features along the X edges are indeed edge states, Jia et al., in their first rebuttal, show $dI/dV(V)$ traces extracted from a conductance map at -70meV . Two comments about these figures reproduced below: while the distance from the edge mentioned in panel b may be correct near the black dotted line, they are not correct for locations near and to the left of the red dotted line which are clearly further away from the edge. Moreover, to find a similar exponential decay (within 26%) of the LDOS with distance from the edge, the authors need to shift the edge position by about 1,2 nm (vertical black lines in panels c and d below) between the two sites located 2,5 nm from each other. These all point at an ill-defined nature of these enhanced features near the step edge.

In summary, the manuscript is rather convincing with respect to the impact of the substrate's dielectric properties on the Y edges, but not with respect to the discussion of the X edges. For this reason, I do not support publication of this manuscript in its present form.

We thank the reviewer for the positive assessment regarding our analysis of the Y edges. Yet, we need to point out there is no difference in our treatment of the X and Y edges, other than that we extract a larger interaction / correlation strength on X edges, reflected in a lower Luttinger parameter.

As it can be challenging to estimate the spatial extend of an exponentially decaying quantity from a colormap, especially in the presence of pronounced charge density modulations, we further illustrate the consistency in edge state analysis below, where we provide additional line cuts, plotting energy-integrated LDOS (panel a), normalized with respect to the LDOS maximum at the edge position ($y=y_0$).

The data indeed confirm a uniform overall enhancement of the edge LDOS, with a pronounced exponential tail into the bulk, despite spatial LDOS modulations. From the data, we extract an edge state decay length of $(2.37 \pm 0.48)\text{nm}$, consistent along the X-edge (panel d), and in very good agreement with that presented in Fig.1 of the main text for the Y-edge ($2.1 \pm 0.2\text{nm}$). The value furthermore agrees with the simple expectation that,

$$\xi = \frac{\hbar v_F}{\Delta_{\text{QSH}}} \sim 2 \text{ nm}$$

Importantly, the consistency of this spatial decay indicates that the edge state is well-defined and is of topological origin in that there is a well-defined imaginary wave vector describing an evanescent wave tail into a well-formed and constant 2D QSH bulk gap Δ_{QSH} . Finally, our data also clearly shows that the maximum of the edge state LDOS (y_0) tracks the position of the physical edge position reasonably well all along the X-edge (panel c).

We would also like to re-emphasize that our conclusions are not drawn from a single data set, but that our study reflects measurements across several tip-sample configurations at over nearly 70 tunnelling points across many different edges. Importantly, the *statistics* of K -values confirm consistent behaviour across the X and the Y edges in terms of universal scaling and dielectric screening, but with the important difference that K is reduced on X edges owing to a reduction in kinetic energy of the carriers.

As noted in our previously revised manuscript and prior responses, in the limit of very strong interactions ($K \sim 1/4$) – reached on the X -edges – additional scattering channels such as two-particle scattering may lead to charge density wave formation (quasi-periodic spatial modulations of the LDOS along the edge state). We would therefore like to point out that the elongated features perpendicular to edge state which the reviewer points at should not be confused with the edge state itself. These features simply arise from charge density modulations superimposed onto an otherwise continuous overall enhancement of the edge's LDOS as shown above.

Further to our previous clarifications on the observation and possible nature of the charge density modulations at the X edges, we have now added further detail in our revised manuscript where it now reads (page 2 last paragraph),

action strength and edge state dispersion. In the limit of very strong interactions ($K < 1/4$) reached in some of our samples, two-particle back-scattering of helical electrons may become significant [4, 24], potentially leading to additional effects such as charge density wave formation, or even fractional Wigner crystallisation [24]. Eq. (1)

On page 3, third paragraph, it now reads,

ference. Superimposed spatial modulations in the charge density can be shown to arise from edge roughness even in otherwise perfect monolayer crystals (Supplementary Fig. 7). However, as these modulations – more clearly pronounced on the X -edges – appear quasi-periodic, additional mechanisms may be at play, including spatial inhomogeneity of the Fermi velocity and/or interaction-mediated scattering of the edge state, leading to charge density wave order.

In the conclusions to the paper, it now reads,

be confirmed in other QSH material systems. In turn, the strongest many-body interactions are found on the unscreened X -edges for WTe_2/BLG where we measure $K = (0.21 \pm 0.01)$ – the lowest reported for any helical TLL system to date. Indeed, this value represents the limit of very strong interactions ($K < 0.25$), where additional mechanisms, such as charge density wave formation due to two-particle scattering or even Wigner crystallisation [24] may be expected. Indeed, the quasi-periodic charge density modulations observed on X -edges are not inconsistent with such phenomena and will be a focus for future investigation into the role of edge disorder and scattering. We thus expect that our results will

Response to Reviewer #3

The authors have performed additional data analysis in response to my question in the last round. The additional experimental curves in the transition region between the X and Y edges, however, look not very well fit to TLL behavior, as also mentioned by the authors. Therefore, I am not fully convinced by their interpretation. At this stage, it might be unfair to stop the manuscript from publication, since there were similar experiments done on other QSH material systems. The experimental findings here are likely to trigger further investigations into this field. I would in principle agree on the publication of the manuscript, although I disagree that the difference in the behavior of X and Y edges is sufficiently investigated and explained.

We thank the reviewer for acknowledging our additional data analysis and for recommending publication at this point. We do acknowledge that the fit quality is limited for two of the points at the corner junction. Despite the somewhat limited energy resolution – owing to the mode of measurement (energy dependent dI/dV maps) in this particular data set, we can see that the TLL fits the pseudogap ZBA quite well for points away from the corner, where K-values are well-defined and in accord with the statistical analysis for the X and Y edge presented in the main text.

The deviations from an ideal TLL model, confined to within $\sim 2\text{nm}$ of the corner junction, may indeed point at the transition / tunnelling between the well-defined X and Y TLL of an otherwise continuous 1D charge distribution. Investigating such phenomena in-dept both from theoretical and experimental perspective, alongside the impact of non-uniformity along X edges and the emergence of charge density modulations, will allow to shed further light on the differences between X and Y edge TLLs as well as the interplay of disorder and 1D electronic correlations in the future.